# CAPO: Cost-Aware Prompt Optimization

**Tom Zehle**[1,*] **Moritz Schlager**[1,*] **Timo Heiß**[1,*] **Matthias Feurer**[1,2]

[1]Department of Statistics, LMU Munich, Munich, Germany
[2]Munich Center for Machine Learning (MCML)
[*]Equal contribution.

**Abstract** Large language models (LLMs) have revolutionized natural language processing by solving a wide range of tasks simply guided by a prompt. Yet their performance is highly sensitive to prompt formulation. While automatic prompt optimization addresses this challenge by finding optimal prompts, current methods require a substantial number of LLM calls and input tokens, making prompt optimization expensive. We introduce CAPO (Cost-Aware Prompt Optimization), an algorithm that enhances prompt optimization efficiency by integrating AutoML techniques. CAPO is an evolutionary approach with LLMs as operators, incorporating racing to save evaluations and multi-objective optimization to balance performance with prompt length. It jointly optimizes instructions and few-shot examples while leveraging task descriptions for improved robustness. Our extensive experiments across diverse datasets and LLMs demonstrate that CAPO outperforms state-of-the-art discrete prompt optimization methods in 11/15 cases with improvements up to 21%p in accuracy. Our algorithm achieves better performances already with smaller budgets, saves evaluations through racing, and decreases average prompt length via a length penalty, making it both cost-efficient and cost-aware. Even without few-shot examples, CAPO outperforms its competitors and generally remains robust to initial prompts. CAPO represents an important step toward making prompt optimization more powerful and accessible by improving cost-efficiency.

## 1 Introduction

The increasing capabilities of transformer-based large language models (LLMs) (Vaswani et al., 2017; Brown et al., 2020) have led to a paradigm shift in Natural Language Processing (NLP): instead of pre-training and expensively fine-tuning models for each individual downstream task, a single LLM, pre-trained in an entirely unsupervised manner, can now solve a diverse range of tasks, simply steered by a textual prompt without requiring any additional training (Liu et al., 2023). These models demonstrate strong performance on many NLP tasks, often nearly reaching performances of state-of-the-art fine-tuned models (Brown et al., 2020). In this context, a prompt refers to instructions provided to the LLM as input to guide its output toward solving a specific task (Karmaker Santu and Feng, 2023; White et al., 2025). It may additionally include in-context examples ("shots") of the task,

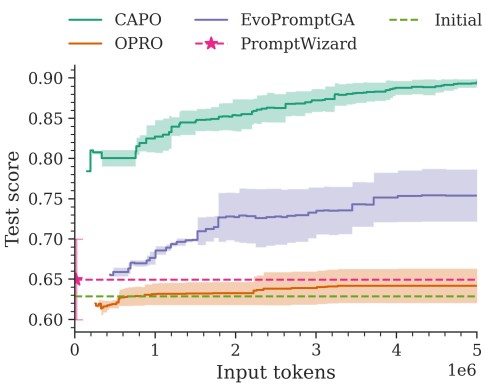

Figure 1: CAPO yields superior mean population test scores on Subj with Qwen2.5-32B.

acting as demonstrations (Brown et al., 2020). However, LLM performance is highly sensitive to prompt quality, format, as well as choice and order of few-shot examples (Zhao et al., 2021; Lu et al., 2022; Zhou et al., 2023). It has been demonstrated that semantically similar prompts can perform quite differently (Yang et al., 2024), which we illustrate in Table 1 with two semantically similar prompts differing by 10%p in accuracy after optimization.

Table 1: Best performing prompts from our benchmark experiments on GSM8K with Llama-3.3-70B.

**Before optimization** (43.8%): Please analyze this elementary school math problem that requires multiple logical steps. After explaining your reasoning, provide the ultimate solution between `<final_answer>` `</final_answer>` tags.

**After optimization with EvoPromptGA** (53.8%): Assist with solving the elementary or grade school level math problem that requires multiple steps and provide the solution within `<final_answer>` `</final_answer>` tags for easy identification.

**After optimization with CAPO** (ours, 79.2%): To tackle this math word problem, which demands a series of logical steps, dissect it methodically. Outline your thought process and ensure you clearly signify your solution, enclosing it within `<final_answer>` `</final_answer>` markers for easy identification. *+ 2 few shots*

This phenomenon introduces the need for prompt engineering or optimization, i.e., designing prompts to enable an LLM to optimally solve a task (Liu et al., 2023; Meskó, 2023). Manual prompt engineering requires time and expertise (Liu et al., 2023). Therefore, automatic prompt optimization has gained increasing attention, including both continuous approaches optimizing learnable "soft prompts" (Lester et al., 2021; Li and Liang, 2021; Qin and Eisner, 2021) and discrete methods acting directly on textual prompts (Zhou et al., 2023; Agarwal et al., 2024; Yang et al., 2024). The discrete prompt optimization framework EvoPrompt (Guo et al., 2024), which leverages LLMs as operators in an evolutionary algorithm, achieves strong performance across various tasks. However, EvoPrompt relies on good, task-specific initial prompts. Other approaches incorporate human-designed task descriptions to mitigate this reliance (Yang et al., 2024). Moreover, recent advances in prompt optimization also integrate few-shot example selection (Agarwal et al., 2024; Wu et al., 2024).

Nonetheless, many prompt optimization methods remain relatively expensive in terms of the number of LLM calls (Agarwal et al., 2024). For instance, optimizing with EvoPrompt in its original parametrization requires 4-6 million input tokens per task until convergence (Guo et al., 2024). Given the API costs for commercial LLMs, this can quickly become expensive: at current commercial API rates, this translates to approximately $34 for GPT-4.1 and $300 for Claude Opus 4[1] for optimization, not even accounting for costs of productively using the optimized prompt.

In this paper, we address the cost problem in prompt optimization by introducing CAPO (Cost-Aware Prompt Optimization), a novel discrete prompt optimization algorithm that integrates AutoML techniques for enhanced cost-efficiency. CAPO draws its underlying mechanism on EvoPrompt (Guo et al., 2024) and combines it with racing (Birattari et al., 2002) to reduce the number of evaluations and improve cost-efficiency. It employs multi-objective optimization by incorporating prompt length as an additional objective through a penalty. In addition, our algorithm integrates recent advances in prompt optimization by combining instruction and few-shot example optimization as well as leveraging task descriptions for improved robustness. Our main contributions are:

1. We introduce CAPO, a cost-efficient and -aware prompt optimization algorithm that integrates racing and multi-objective optimization, leveraging few-shot examples and task descriptions.
2. We conduct extensive benchmark experiments comparing CAPO against three state-of-the-art prompt optimization algorithms across diverse datasets and LLMs, demonstrating its superior performance in most scenarios, even with substantially fewer input tokens (see, e.g., Figure 1).
3. We provide comprehensive ablation studies indicating that few-shot example selection greatly enhances performance, racing improves cost-efficiency, the prompt length objective reduces average prompt length, and task descriptions make the algorithm robust to initial prompt quality.

We make our complete implementation publicly available under the Apache 2.0 license at `https://github.com/finitearth/capo/` to facilitate reproducibility and adoption.

---

[1]Depending on the task, 2-3 million output tokens are additionally required. Considering input token costs of $2 / 1M tokens for GPT-4.1 ($15 / 1M tokens for Claude Opus 4) and output token costs of $8 / 1M tokens ($75 / 1M tokens), we arrive at a total of roughly $34 ($300). For API prices, see `https://openai.com/api/pricing/` and `https://www.anthropic.com/pricing` (accessed: 2025-03-22).

## 2 Notation & Problem Statement

Let $\mathcal{I}$ denote the space of all possible instructions $i$ and $\mathcal{E}$ the space of all possible examples $e$, also referred to as "shots". A tuple of few-shot examples consisting of $k$ shots is denoted by $\boldsymbol{e} = (e_1, \ldots, e_k)$, the space of all possible $k$-shot examples is represented by $\mathcal{E}^k$. We define the space of possible prompts with up to $k_{\max}$ shots as $\mathcal{P} = \mathcal{I} \times \bigcup_{k=0}^{k_{\max}} \mathcal{E}^k$, where each prompt $p = (i, \boldsymbol{e})$ consists of an instruction and between 0 and $k_{\max}$ shots. Let an LLM be a function $\Phi$ that takes a prompt $p$ and some input, and produces an output. In the classical case, the input refers to an instance $x \in \mathcal{X}$ from a dataset $\mathcal{D} = \{(x^{(i)}, y^{(i)})\}_{i=1}^n \sim \mathbb{P}_{xy}$ and the output to a corresponding predicted label $\hat{y} \in \mathcal{Y}$. We also use LLMs for generating variations of instructions, where input and output both refer to instructions $i$. We refer to this as meta-LLM in contrast to the evaluation-LLM for which we optimize the prompt. LLMs are treated as black boxes without access to gradients or token probabilities, a common scenario for API LLMs from closed-source vendors.

We evaluate a prompt $p$ by comparing the true label $y$ to the predicted label $\hat{y} = \Phi(p, x)$ for a given instance $x$ with a point-wise scoring function $\sigma : \mathcal{Y} \times \mathcal{Y} \to \mathbb{R}$. While any scoring function is generally possible, we always test for direct match using

$$\sigma(y, \hat{y}) = \begin{cases} 1 & \text{if } y = \hat{y} \\ 0 & \text{otherwise} \end{cases} \tag{1}$$

as scoring function. Our goal is to find a prompt $p$ that maximizes this score in expectation:

$$\arg\max_{p \in \mathcal{P}} \mathbb{E}_{(x,y) \sim \mathbb{P}_{xy}}[\sigma(y, \Phi(p, x))]. \tag{2}$$

Estimating this quantity based on a finite dataset $\mathcal{D} = \{(x^{(i)}, y^{(i)})\}_{i=1}^n$ yields our objective $f$: $f(p; \mathcal{D}) = \frac{1}{n} \sum_{i=1}^n \sigma(y_i, \Phi(p, x_i))$. Our goal is to find a prompt $p$ that maximizes $f$ within a limited budget of input tokens to an LLM. Since we want to generalize well to unseen data, we measure $f$ on a separate, finite test dataset $\mathcal{D}_{test} = \{(x^{(i)}, y^{(i)})\}_{i=n+1}^{n+m}$ drawn from the same distribution.

## 3 Related Work

**Automatic Prompt Optimization**. Recently, interest in automating prompt optimization has grown as manual prompt engineering requires time and expertise without guaranteeing optimality (Jiang et al., 2020; Liu et al., 2023). A related area is prompt selection, which aims to find optimal prompts from a pre-defined pool of candidates (Sorensen et al., 2022; Do et al., 2024; Schneider et al., 2024; Shi et al., 2024). Prompt optimization includes both the optimization of instructions and the selection of relevant few-shot examples ("exemplar optimization") (Wan et al., 2024; Wu et al., 2024).

Continuous prompt optimization improves prompts in continuous space to obtain learnable "soft prompts" (Li and Liang, 2021; Lester et al., 2021; Qin and Eisner, 2021). While this requires access to LLM parameters and makes prompts not interpretable (Lester et al., 2021), recent approaches like InstructZero (Chen et al., 2024) and its extension INSTINCT (Lin et al., 2024) address this by performing Bayesian optimization on soft prompts used to generate human-readable instructions.

Discrete methods directly optimize textual prompts (Agarwal et al., 2024). Unlike earlier approaches that require access to gradients or token probabilities (Shin et al., 2020; Deng et al., 2022; Shi et al., 2023), recent discrete methods also work with black box LLMs. They typically use a "meta-LLM" instructed by a "meta-prompt" to alternate prompt candidates: APE (Zhou et al., 2023) uses a meta-LLM to generate instructions from demonstrations and iteratively proposes semantically similar variants, ProTeGi (Pryzant et al., 2023) leverages mispredicted instances as "pseudo-gradients", and PromptBreeder (Fernando et al., 2024) uses an evolutionary strategy with a meta-LLM guided by self-improving mutation-prompts. EvoPrompt (Guo et al., 2024), which serves as foundation of our work, is also based on evolutionary algorithms and has two instantiations: a

genetic algorithm (GA) and differential evolution (DE). Both implement evolutionary operations by a meta-LLM. Despite outperforming previous discrete methods, EvoPrompt has two major drawbacks: it requires many LLM calls (Agarwal et al., 2024) and its performance depends on a good, task-specific initial prompt population (Yang et al., 2024). OPRO (Yang et al., 2024) directly employs LLMs as optimizers by leveraging task descriptions, task examples, and previous candidates with scores in the meta-prompt, maintaining good performance even with task-unspecific initial prompts. These methods optimize instructions without incorporating few-shot examples in prompt candidates. However, even simple random example selection can outperform sophisticated instruction optimizers. Combining instruction and example optimization is found to create synergies (Wan et al., 2024). PromptWizard (Agarwal et al., 2024) optimizes instructions and examples simultaneously using a critique-synthesis mechanism, reportedly outperforming previously described methods while greatly reducing LLM calls. However, approaches like PromptWizard, ProTeGi, or OPRO require a notion of what constitutes a "good" prompt, asking a meta-LLM to identify problems or improve prompts. Since prompt performance does not necessarily follow predictable patterns (Yang et al., 2024), this potentially limits these methods' ability to capture such subtleties.

While the methods described above focus on refining single prompts, there are frameworks that treat black-box prompt optimization as a component of larger dynamic systems. TextGrad (Yuksekgonul et al., 2025) introduces a pipeline-based optimization approach that uses LLM-generated feedback as textual gradients to improve prompts. DSPy (Khattab et al., 2024) offers a modular programming framework for building LLM pipelines with an embedded optimization component. Aviary (Narayanan et al., 2024) reframes prompt optimization within the broader context of language agents solving complex tasks through a partially observable multi-step decision process.

**AutoML for Efficiency**. The field of AutoML offers several techniques to enhance optimization efficiency and methods like multi-fidelity optimization (Jamieson and Talwalkar, 2016; Li et al., 2018; Falkner et al., 2018; Awad et al., 2021) have also been successfully adopted outside AutoML, e.g., for prompt selection, where efficiency is similarly important (Schneider et al., 2024; Shi et al., 2024). Racing algorithms are applicable when objectives are decomposable into cheaper sub-objectives that can be evaluated individually. They sequentially evaluate candidates and eliminate poor ones once sufficient statistical evidence accumulates, preserving budget for promising candidates (Birattari et al., 2002, 2010). Important works include Hoeffding Races (Maron and Moore, 1994) using Hoeffding's bound for elimination, BRACE (Moore and Lee, 1994) employing Bayesian statistics, F-Race (Birattari et al., 2002) using Friedman's test (Conover, 1999), and I/F-Race (Balaprakash et al., 2007) iteratively applying F-Race while biasing a probabilistic model of the candidates to promising areas. The *irace* package (López-Ibáñez et al., 2016) provides a general iterated racing implementation, including a paired t-test as alternative. Related methods that save evaluations by adaptively increasing evaluations include FocusedILS (Hutter et al., 2009), as well as ROAR and SMAC (Hutter et al., 2011), employing an "intensification" mechanism without statistical testing.

Multi-objective optimization (MOO) addresses scenarios with multiple competing objectives such as performance versus efficiency (Karl et al., 2023). A priori methods transform multiple objectives into a single one, e.g., via scalarization, yielding only a single solution candidate (Karl et al., 2023). While greatly simplifying optimization (Miettinen, 1998), choosing scalarization weights a-priori is often non-trivial (Jin and Sendhoff, 2008). A posteriori methods produce a set of Pareto-optimal solutions (Karl et al., 2023). Notable approaches include evolutionary methods like NSGA-II (Deb et al., 2002) and SMS-EMOA (Beume et al., 2007) based on non-dominated sorting rank, and Bayesian optimization approaches such as ParEGO (Knowles, 2006), approximating the Pareto-front using a set of randomly generated scalarization weights. Combinations of MOO and racing include irace with Hypervolume (López-Ibáñez et al., 2016), S-Race and its extensions (Zhang et al., 2013, 2015a; Miranda et al., 2015), MO-ParamILS (Blot et al., 2016), and MO-SMAC (Rook et al., 2025).

For additional background on the discussed approaches, we refer to Appendix A.

## 4 CAPO: Cost-Aware Prompt Optimization

We introduce CAPO (Cost-Aware Prompt Optimization), a discrete prompt optimization algorithm that addresses the cost problem in automatic prompt optimization and integrates recent prompt optimization advances. Conceptually, CAPO builds on EvoPromptGA (Guo et al., 2024), following a standard genetic algorithm (Goldberg, 1989) with a meta-LLM for cross-over and mutation operations. As the number of evaluations is a major cost factor in prompt optimization, CAPO employs racing to eliminate underperforming candidates early. In addition, CAPO draws inspiration from multi-objective optimization, incorporating efficiency as additional objective by penalizing prompt length. Keeping the length of the resulting prompt minimal reduces evaluation cost during optimization and deployment cost of the final prompt. Similar to PromptWizard (Agarwal et al., 2024), CAPO optimizes both instructions and few-shot examples simultaneously. Furthermore, CAPO leverages task descriptions in the meta-prompt to reduce reliance on task-specific initial prompts (Yang et al., 2024). We additionally simplify the meta-prompt templates by substantially shortening them and avoiding formulations like "better prompt" that require a notion of what constitutes a good prompt. We now describe the CAPO algorithm as outlined in Algorithm 1.

**Population Initialization**: A set of initial instructions $\mathcal{I}_0$ of population size $\mu$ is provided as input, either manually engineered or automatically generated with approaches like APE (Zhou et al., 2023). We first augment each instruction with a random number of few-shot examples between 0 and $k_{max}$. For each example, we generate reasoning to provide richer information compared to solely a label as example output. We prompt the evaluation-LLM with the initial instruction to solve the example input, which typically yields a response with both reasoning and prediction. If the LLM fails to generate a correct prediction, we use the true label as example output.[2] This resembles PromptWizard (Agarwal et al., 2024), which leverages reasoning chains. This initialization procedure yields a diverse population with varying number and lengths of shots.

**Cross-over & Mutation**: For cross-over, CAPO randomly selects parents, unlike Evo-PromptGA (Guo et al., 2024), which uses score-based roulette wheel selection. While less exploitative, our choice eliminates expensive evaluations during parent selection. The CROSS_OVER operation (cf. Appendix B) leverages a meta-LLM $\Phi_{meta}$ to create an offspring instruction $i_{off}$ from

---

**Algorithm 1** CAPO: Cost-Aware Prompt Optimization

---

**Require:** datasets $\mathcal{D}_{dev}$ and $\mathcal{D}_{shots}$, meta-LLM $\Phi_{meta}$, evaluation-LLM $\Phi_{eval}$, initial instructions $\mathcal{I}_0 = \{i_1, \ldots, i_\mu\}$, population size $\mu$, block size $b$, number of iterations $T$, number of crossovers per iteration $c$, max. number of few-shot examples $k_{max}$, max. number of evaluated blocks $z_{max}$, confidence level $\alpha$, token length penalty control parameter $\gamma$, cross-over-meta-prompt $p_C$, mutation-meta-prompt $p_M$
1: Divide dataset $\mathcal{D}_{dev}$ into blocks $\mathcal{B} = \{B_1, ..., B_z\}$ where $|B_i| = b$
2: $\mathcal{P}_\mu \leftarrow []$
3: **for** $i \in \mathcal{I}_0$ **do**                                                  ▷ Initialize prompt population
4:     $k \sim \text{Unif}(\{0, \ldots, k_{max}\})$                                      ▷ Sample number of few-shots
5:     $e \leftarrow \text{CREATE\_SHOTS}(\mathcal{D}_{shots}, k, i, \Phi_{eval})$              ▷ Create few-shots
6:     $p \leftarrow (i, e)$
7:     $\mathcal{P}_\mu \leftarrow \text{APPEND}(p, \mathcal{P}_\mu)$
8: **end for**
9: **for** $t = 1$ to $T$ **do**
10:     $\mathcal{P}_{off} \leftarrow \text{CROSS\_OVER}(\mathcal{P}_\mu, \Phi_{meta}, p_C, c)$         ▷ Perform cross-over operation
11:     $\mathcal{P}_{off} \leftarrow \text{MUTATE}(\mathcal{P}_{off}, \Phi_{meta}, \Phi_{eval}, p_M, \mathcal{D}_{shots}, k_{max})$   ▷ Mutation operation on offspring
12:     $\mathcal{P}_\mu \leftarrow \text{DO\_RACING}(\mathcal{P}_\mu \cup \mathcal{P}_{off}, \mathcal{B}, \Phi_{eval}, \alpha, \gamma, \mu, z_{max})$   ▷ Survival selection via racing
13: **end for**
14: **return** $\mathcal{P}_\mu$

---

[2] For illustration purposes, we provide exemplary few-shot examples with and without reasoning in Appendix E.2.

the two selected parents' instructions. The meta-LLM is steered by a meta-cross-over prompt $p_C$, which is simplified compared to the EvoPromptGA meta-prompt (Guo et al., 2024) and incorporates a task description.[3] For the offspring's few-shot examples $e_{\text{off}}$, we sample from the union of the parents' examples, with the number of examples corresponding to the average of the number of few-shot examples of the two parents. This process is repeated $c$ times per iteration to generate $c$ offspring. To each offspring, we then apply the MUTATE operation (cf. Appendix B). Similar to cross-over, a meta-LLM $\Phi_{\text{meta}}$ is instructed via a simplified meta-mutation-prompt $p_M$ with task description to create a mutated version of the offspring instruction.[3] To mutate few-shot examples, we apply one of three operations with equal probability: adding a new shot if not exceeding $k_{\text{max}}$, removing a random shot if there are any, or keeping them unchanged. Following this step, we randomly shuffle the order of all examples. This approach of modifying only single examples and their order is designed to foster local exploration of the few-shot example choice and their quantity.

**Survival Selection**: To select survivors, we eliminate prompts through racing (DO_RACING, cf. Appendix B), discarding underperforming prompts early when statistical evidence indicates they perform significantly worse. Our racing procedure operates on blocks of samples $\mathcal{B} = \{B_1, ..., B_z\}$ of fixed size $b$, similar to F-Race (Birattari et al., 2002). We optionally shuffle block order in each iteration to avoid potential elimination biases. We sequentially process blocks, evaluate all prompts on the selected block (caching block scores to save evaluations later), and eliminate inferior prompts when more than $\mu$ other prompts are significantly better according to a statistical test. We do not correct for multiple testing as this can negatively affect racing behavior by making the test more conservative, leading to fewer early eliminations of candidates (Birattari, 2009). This corresponds to a population-based racing approach since we compare across the entire population rather than against a single incumbent.[4] Racing continues with additional blocks until we either reach $\mu$ survivors or the maximum block evaluation limit $z_{\text{max}}$. If more than $\mu$ prompts survive after $z_{\text{max}}$ evaluated blocks, we select the $\mu$ best-performing prompts based on their average scores.

As statistical test, we employ a paired t-test with $\alpha = 0.2$, which is favorable for our case compared to the commonly used F-test as scores across instances are commensurable (López-Ibáñez et al., 2016) while less conservative than non-parametric bounds like Hoeffding's (Maron and Moore, 1994). Since the paired t-test requires normality or sufficiently large sample sizes ($\geq 30$) (Hsu and Lachenbruch, 2014), block size $b$ must be chosen such that assumptions hold even for a single block.

Since we aim to maximize performance while keeping prompt length minimal, i.e., shorter instructions, fewer examples, and reasoning only when necessary, we implement a form of multi-objective optimization. This is particularly important given our inclusion of few-shot examples, which can considerably increase the prompt length. To keep the racing procedure simple, we scalarize our objective using a length penalty parameter $\gamma$ that controls the trade-off between prompt performance and any measure of relative token length. This parameter must be selected a-priori, yielding the objective $f_\gamma(p; B) = f(p; B) - \gamma \cdot \text{REL\_TOKEN\_LENGTH}(p)$. In our implementation, REL_TOKEN_LENGTH represents token count[5] normalized by the longest initial prompt.

## 5 Experimental Setup

For our experiments, we use three different LLMs: *Llama-3.3-70B-Instruct-GPTQ* (Meta, 2024), *Qwen2.5-32B-Instruct-GPTQ* (Qwen: Yang et al., 2025) and *Mistral-Small-24B-GPTQ* (Mistral AI Team, 2025). These cover different model sizes from different companies and regions. We opt for model sizes that still fit on a single GPU while exhibiting strong performances. To meet hardware constraints, we employ GPTQ-quantized models (Frantar et al., 2023), which show negligible

---

[3]Prompt templates are provided in Appendix D.3. We illustrate instruction variation with examples in Appendix E.1.

[4]This makes the erroneous elimination of the best candidate very unlikely, as not only one but several type I errors would have to occur.

[5]We give details on how we count tokens in this paper in Appendix C.4.

performance loss compared to uncompressed models. For each setup, we use the same model as meta- and evaluation-LLM. For further technical details, we refer to Appendix C.

We employ five datasets spanning a diverse range of typical NLP tasks with different subject areas, targets, and complexity levels: *SST-5* (sentiment classification; Socher et al., 2013), *AG News* (topic classification; Zhang et al., 2015b), *Subj* (subjectivity classification; Pang and Lee, 2004), *GSM8K* (grade school math word problems; Cobbe et al., 2021) and *(Balanced) COPA* (commonsense causal reasoning; Kavumba et al., 2019). The first three datasets are used in the EvoPrompt paper (Guo et al., 2024), GSM8K in OPRO (Yang et al., 2024) and PromptWizard (Agarwal et al., 2024), and COPA is added as, to the best of our knowledge, a novel application for discrete prompt optimization. For each dataset, we use 200 samples as few-shot dataset, 300 as development set for optimization (larger than EvoPrompt (Guo et al., 2024), where 200 samples are used for these tasks), and 500 holdout samples as test set (equivalent to the size of the smallest test set from our five datasets; details in Appendix C.2). We automatically create a diverse pool of 15 initial instructions per dataset with Anthropic's Claude Sonnet 3.7 (cf. Appendix D.2), and sample the initial instructions from this pool for all optimizers and models. CAPO and OPRO (Yang et al., 2024) additionally use task descriptions, which we manually craft (cf. Appendix D.1).

We benchmark CAPO against three state-of-the-art discrete prompt optimizers: Evo-PromptGA (Guo et al., 2024), OPRO (Yang et al., 2024), and PromptWizard (Agarwal et al., 2024). We use the GA instantiation of EvoPrompt as it performs similar to the DE variant while being conceptually simpler and closer to CAPO. For EvoPromptGA and OPRO, we use reimplementations of a public library while using PromptWizard's original implementation with small adaptions. For implementation and parametrization details of the optimizers, we refer to Appendix C.4.

For all experiments with CAPO, EvoPromptGA, and OPRO, we do not restrict maximum iterations but instead use a budget of 5M input tokens after which the run terminates.[6] We choose this budget such that EvoPromptGA, which is most expensive in terms of LLM calls, has likely converged (cf. Guo et al., 2024). We evaluate each optimizer with each LLM and dataset, performing three repetitions with different random seeds per setup to quantify variance.

## 6 Results & Analysis

### 6.1 Benchmark Results

We report the test scores of our benchmark experiments in Table 2. The results demonstrate that CAPO outperforms the other prompt optimization methods on most datasets and models (11/15). Notably, for Llama-3.3-70B, CAPO leads to the best results on every single dataset. For scenarios in which another optimizer is better, CAPO is still competitive and within one standard deviation. While performance gains of CAPO compared to the rest are small on SST-5 or AG News, we observe substantial performance improvements on Subj and GSM8K, with up to 21%p improvement over the rest (Llama-3.3-70B on GSM8K). Initial instructions are consistently improved by CAPO.

To assess the performance at intermediate token budgets, we depict the mean population performance over input tokens for two representative examples of optimizer-dataset pairs in Figures 1 & 2 (see Appendix H.2 for the remaining optimization curves). For both examples, as soon as CAPO yields the first prompt, it consistently dominates the other optimizers over the entire token range. Early performances of CAPO already exceed the other optimizers' final performances after the full budget, underscoring its cost-efficiency. However, we observe that CAPO often yields its first prompt later in terms of used input tokens than its competitors. This is due to the fact that CAPO includes few-shot examples, making evaluations more costly. It follows that CAPO requires many tokens in the first step while being very cost-efficient later (see Appendix I.2 for details).

---

[6]PromptWizard has no clear way to increase compute time, we report its performance on reduced budget (for details, see Appendix C.4).

Table 2: Performance comparison of different prompt optimizers (last step before exceeding 5M input tokens). We report the mean accuracy (in %) on test set with standard deviation across three seeds of the best prompts. The best prompt per seed is selected from the last population based on development set scores. Bold values indicate best, underlined values second to best performance for each LLM and dataset.

| Model | Optimizer | SST-5 | AG News | Subj | GSM8K | COPA | Avg. |
|---|---|---|---|---|---|---|---|
| Llama-3.3-70B | Initial | 58.47± 1.53 | 87.06± 0.65 | 62.00±5.22 | 44.28± 4.91 | 97.65± 1.31 | 69.89 |
| | OPRO | 60.87± 1.09 | 88.20± 0.49 | 71.33±2.80 | 51.87± 2.04 | 98.07± 0.57 | 74.07 |
| | PromptWizard | 32.80± 1.73 | 23.33± 0.19 | 51.93±0.25 | 39.33±15.09 | 50.33± 0.34 | 39.55 |
| | EvoPromptGA | 60.53± 1.73 | 88.67± 0.41 | 75.53±1.39 | 50.87± 0.74 | 97.60± 1.13 | 74.64 |
| | CAPO (ours) | 62.27± 0.34 | 88.80± 0.75 | 91.60±2.16 | 73.73± 3.73 | 98.27± 0.52 | 82.93 |
| Qwen2.5-32B | Initial | 56.68± 1.94 | 79.57± 0.84 | 62.85±4.53 | 33.08± 7.78 | 98.27± 0.43 | 66.09 |
| | OPRO | 57.00± 0.43 | 79.87± 0.19 | 70.67±2.36 | 46.33± 3.07 | 98.67± 0.34 | 70.51 |
| | PromptWizard | 39.73±12.31 | 63.47±28.49 | 64.93±5.01 | 15.27±20.19 | 98.13± 0.19 | 56.31 |
| | EvoPromptGA | 58.60± 1.73 | 81.73± 1.68 | 75.87±3.58 | 61.27± 8.39 | 97.87± 0.66 | 75.07 |
| | CAPO (ours) | 59.07± 0.50 | 87.07± 0.81 | 91.00±0.65 | 60.20± 4.82 | 98.47± 0.19 | 79.16 |
| Mistral-Small-24B | Initial | 48.69± 2.94 | 72.21± 7.45 | 61.65±6.04 | 33.71± 5.89 | 94.56± 0.94 | 62.17 |
| | OPRO | 53.20± 2.83 | 84.20± 0.16 | 77.07±0.09 | 43.53± 0.47 | 96.33± 0.34 | 70.87 |
| | PromptWizard | 31.07± 3.80 | 44.40±25.76 | 59.00±5.09 | 48.67± 6.46 | 57.47±10.28 | 48.12 |
| | EvoPromptGA | 54.93± 0.94 | 84.40± 0.28 | 74.93±2.04 | 43.93± 3.85 | 96.13± 0.34 | 70.87 |
| | CAPO (ours) | 60.20± 0.33 | 84.33± 2.13 | 81.67±1.64 | 65.07± 1.20 | 95.13± 1.20 | 77.28 |

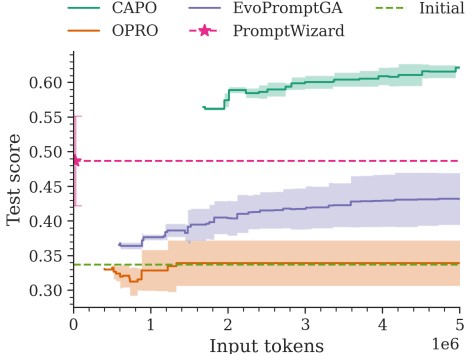

Figure 2: Population mean test scores over input tokens on GSM8K with Mistral-Small-24B with mean ± std across seeds. PromptWizard yields only a single prompt early, marked with a star and error bars.

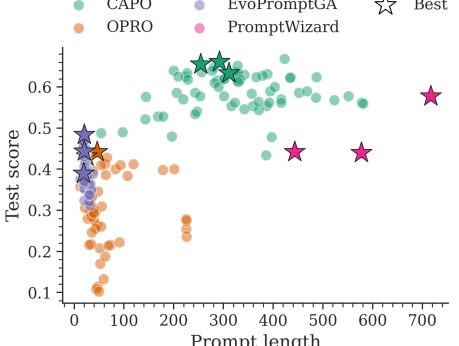

Figure 3: Test scores vs. prompt length (system + user prompt) for every prompt on GSM8K with Mistral-Small-24B. Stars mark the best performing on dev-set from the last population.

We also find that CAPO yields longer prompts than EvoPromptGA and OPRO due to few-shot examples but still shorter than PromptWizard (cf. Figure 3). Thus, though PromptWizard requires fewer tokens during optimization (on average only 25k input tokens), CAPO reduces the deployment cost of the optimized prompts. A more detailed analysis of CAPO's prompt length reveals that, on average, 66% of the tokens can be attributed to few-shot examples compared to the instruction, with up to 92% in the prompts of Llama-3.3-70B on GSM8K (cf. Appendix G.2).

Further, we identify the evaluation of prompts as the main driver behind the token usage: on average, 97% of the input tokens are consumed by the evaluation-LLM and only 3% by the meta-LLM (cf. Appendix G.1), justifying our approach of reducing evaluation costs through racing.

## 6.2 Ablation Studies

To better understand design choices in CAPO, we ablate several components on AG News and GSM8K with Llama-3.3-70B, a budget of 5M input tokens, three seeds, and optimizer parameters as

before. We provide results in Table 3 and give further insights in Appendix I while describing the key findings here.

**I. Zero-shot performance:** Without few-shot examples, the performances of the best prompts remain unchanged for AG News while being substantially worse for the more complex GSM8K task (cf. Table 3). This highlights the importance of few-shot examples for complex tasks. Notably, zero-shot CAPO still considerably outperforms EvoPromptGA on GSM8K. Due to the lack of few-shot examples, the resulting prompts are much shorter than default CAPO prompts but interestingly longer than for EvoPromptGA. We find that this is due to our meta-prompt template sometimes causing repetitions inside optimized prompts after cross-over.[7]

Table 3: Ablation study results using Llama-3.3-70B. Mean accuracy (in %) on test set of best prompt per seed selected on the development set scores (format as in Table 2). "all above" is the combination of zero-shot, $\gamma = 0$, and w/o racing.

| Ablation | Accuracy | | Prompt length | |
|---|---|---|---|---|
| | AG News | GSM8K | AG News | GSM8K |
| CAPO | 88.80±0.75 | 73.73±3.73 | 481±113 | 110±46 |
| ↪ *zero-shot* | 89.00±0.16 | 59.20±5.03 | 94± 17 | 74±24 |
| ↪ *γ = 0* | 89.27±0.41 | 74.93±1.04 | 297± 27 | 128±27 |
| ↪ *w/o racing* | 89.20±0.43 | 75.00±3.12 | 469±130 | 146±52 |
| ↪ *all above* | 88.53±0.09 | 50.93±5.25 | 78± 11 | 37±13 |
| ↪ *generic init* | **89.33**±0.19 | **82.93**±2.36 | 206±113 | 182±22 |
| EvoPromptGA | 88.67±0.41 | 50.87±0.74 | 28± 2 | 30± 1 |
| ↪ *generic init* | 23.20±0.00 | 53.47±0.38 | **17**± 8 | **20**± 2 |

**II. No length penalty:** Removing the length penalty ($\gamma = 0$) improves performance of the final prompts compared to default CAPO while the prompt length stays in a similar range (cf. Table 3). Such a performance improvement is expected as disabling the length penalty results in directly optimizing accuracy. Nonetheless, we find that with length penalty, average prompt length decreases as optimization progresses, enabling more steps. We discuss this effect of different length penalties in Appendix F.

**III. No racing:** After 5M input tokens, CAPO without racing performs slightly better while differences lie within one standard deviation (cf. Table 3). Still, comparing performance over input tokens reveals that with racing, substantially fewer input tokens are needed to yield first prompts with relatively good performance (cf. Figure 16). We further find that racing, on average, saves 44% of evaluations, enabling considerably more steps with the same budget (cf. Appendix I.2).

**IV. No shots, length penalty & racing:** To identify the joint influence of our core innovations, we combine the three ablations above by removing few-shot examples, racing, and the length penalty. The resulting CAPO configuration shows a considerable performance drop compared to the default CAPO on GSM8K while only slightly losing performance on AG News. Generally, we observe that, as expected, this CAPO configuration yields performances very similar to EvoPromptGA. Thus, the performance gains of default CAPO over EvoPromptGA must stem from our core innovations and their interplay.

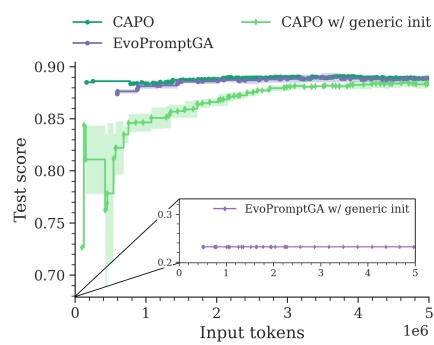

Figure 4: Population mean test scores over input tokens on AG News with Llama-3.3-70B with mean ± std across seeds. CAPO and EvoPromptGA start from task-specific initial instructions, their respective counterparts from generic initial instructions.

**V. Generic initial instructions:** We use automatically generated task-unspecific initial instructions (cf. Appendix D.2) and analyze if task descriptions in CAPO counteract degrad-

---

[7]The cross-over step prompts the meta-LLM to "[...] merge the two prompts [...]" (cf. Appendix D.3), which leads to occasional concatenation of prompts.

ing performances observed by Yang et al. (2024). Our results confirm the degrading performance of EvoPromptGA, especially for AG News (cf. Figure 4). The optimization curves reveal that EvoPromptGA's performance stays constant as no valid labels are predicted while CAPO starts lower than with task-specific instructions but quickly improves as task descriptions introduce task-specific information, eventually reaching similar performances. Surprisingly, for GSM8K, generic initial instructions even lead to improved CAPO performance (cf. Table 3), likely because (1) instances of GSM8K contain instructions themselves and (2) CAPO can explore more freely. This demonstrates CAPO's robustness and suggests even generic instruction repositories could serve as initial populations.

## 7 Conclusion & Future Work

In this paper, we propose the discrete prompt optimization method CAPO, an evolutionary algorithm that integrates racing and multi-objective optimization, leveraging few-shot examples and task descriptions. Our experiments demonstrate that CAPO outperforms other discrete prompt optimizers in 11 out of 15 cases, with differences up to 21%p on GSM8K with Llama-3.3-70B, while being competitive in the remaining 4 cases. CAPO yields better performance already at earlier stages than other algorithms after the full budget, showing its cost-efficiency, and remains dominant over the entire budget. Nonetheless, it yields longer prompts due to few-shot examples. Our ablation studies reveal several important insights: (I.) few-shot examples substantially contribute to the performance, especially for complex tasks, while CAPO maintains strong performance even without examples; (II.) the length-penalty effectively reduces average prompt length throughout optimization; (III.) racing leads to considerable savings in terms of evaluations, enabling more iterations; (IV.) CAPO's performance gains must be due to few-shot examples, length penalty, and racing, as well as their interplay; and (V.) task descriptions make CAPO robust, yielding strong performance with generic initial instructions.

Despite the great advances, our work also has limitations. First, while racing reduces evaluations, it does not necessarily contribute to better performance after the full budget. Moreover, our study focuses on smaller models and could thus be extended to larger LLMs. Similarly, our analysis is limited to classification and math tasks while the main usage of LLMs is text generation. Additionally, all datasets are older than the LLMs, leading to potential test set contamination. Nonetheless, this limitation holds for all optimizers equally, not affecting our conclusions. Finally, output token length is another major cost factor influenced by the prompt, which is not considered in our work and should be addressed by future work.

In the future, we plan to make CAPO an a posteriori multi-objective method, allowing the user to choose from a final population that features different trade-offs in prompt performance and length. In addition, we plan to study the use of other strategies for budget allocation, such as successive halving (Karnin et al., 2013; Parmentier et al., 2019) or hyperband (Li et al., 2018; Awad et al., 2021).

## 8 Broader Impact Statement

Making CAPO openly available enables positive impacts across industrial and research applications, though also creating potential for misuse by malicious actors. As our work builds upon LLMs, it inherits their associated impacts, including potential biases, hallucination, and energy consumption. Prompt optimization specifically requires numerous LLM calls, resulting in significant energy expenditure and negative environmental impact. Nonetheless, CAPO aims to reduce these costs. Through racing, CAPO saves evaluations while producing effective prompts earlier. A length penalty encourages shorter prompts for reduced production costs. Our algorithm often achieves better performance at a substantially smaller input token budget than other optimizers on the full budget, greatly improving cost-efficiency. These efficiency improvements directly translate to reduced energy requirements for more environmentally sustainable prompt optimization.

**Acknowledgements.** We would like to thank Lennart Schneider for his invaluable suggestions and impulses through multiple discussions. We also gratefully acknowledge the computational and data resources provided by the Leibniz Supercomputing Centre.

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

# Appendix

## A Background

This section provides additional background on concepts and algorithms employed in this paper.

### A.1 Prompt Optimization Algorithms

In the following, we present the three prompt optimization algorithms we benchmark against: EvoPrompt (Guo et al., 2024), OPRO (Yang et al., 2024), and PromptWizard (Agarwal et al., 2024). All three are of discrete prompt optimization methods. They optimize textual prompts directly by generating multiple prompt variations and selecting the best candidates. Thus, they do not require access to gradients or token probabilities but are also applicable to black box LLMs.

**EvoPrompt**. EvoPrompt (Guo et al., 2024) is a discrete prompt optimization framework based on evolutionary algorithms. It uses a meta-LLM to alternate prompts via cross-over and mutation operations, enabling direct optimization of discrete prompts while maintaining coherence and human readability. EvoPrompt starts from an initial prompt population, iteratively generates new prompts, evaluates generated candidates on a development set, selects the best performing ones as survivors, and terminates after a predefined number of iterations.

Guo et al. (2024) present two instantiations of EvoPrompt: as a Genetic Algorithm (GA) and as a Differential Evolution (DE) method. We focus on EvoPromptGA, which serves as basis for our algorithm and is therefore also used in our benchmark experiments. In each iteration, EvoPromptGA selects two parent prompts via roulette wheel selection and generates new candidate prompts in two steps: first, the cross-over operation combines properties from both parents into an offspring; second, each offspring is mutated through small random modifications. Both evolutionary operations are implemented through a single meta-prompt instructing the meta-LLM (for the template, see Appendix D.3). Each iteration produces $\mu$ new prompts that compete with the existing $\mu$ ones, from which the top $\mu$ survive.

Experiments by Guo et al. (2024) across language understanding, generation, and BIG-Bench Hard (BBH) tasks demonstrate that both EvoPrompt instantiations outperform human-written instructions and previous prompt optimizers such as APE (Zhou et al., 2023) and APO/ProTeGi (Pryzant et al., 2023). This makes EvoPrompt a suitable reference for our benchmark experiments.

However, EvoPrompt has two major drawbacks: First, as pointed out in the main paper, it is cost-intensive: it requires a total of $\mu \cdot T \cdot (1 + |\mathcal{D}_{\mathrm{dev}}|)$ LLM calls (Guo et al., 2024) with population size $\mu$, number of iterations $T$, and development set size $|\mathcal{D}_{\mathrm{dev}}|$. This number is mainly driven by the size of $\mathcal{D}_{\mathrm{dev}}$, which is usually much larger than $\mu$ and $T$. Second, as identified by Yang et al. (2024), EvoPrompt's performance can degrade with poor or task-unspecific prompts due to its reliance on task specification via prompt population, a phenomenon we also demonstrate in the main paper. Both shortcomings of EvoPrompt are addressed by CAPO.

**OPRO**. OPRO (Optimization by PROmpting) (Yang et al., 2024) directly employs LLMs as optimizers by specifying optimization tasks in natural language. When used for prompt optimization, a meta-LLM generates new prompt candidates at each iteration, guided by a meta-prompt that contains the task description, task examples, and previously generated candidates with their scores. New candidates are evaluated and appended to the meta-prompt for the subsequent iteration. This approach substantially outperforms human-designed prompts on GSM8K and BBH tasks. Unlike EvoPrompt, OPRO maintains good performance even with task-unspecific initial instructions by leveraging explicit task descriptions and examples within the meta-prompt.

However, OPRO, similar to EvoPrompt, focuses solely on instruction optimization without incorporating few-shot examples (those are only used in the meta-prompt), despite evidence that such examples can significantly improve LLM performance (Brown et al., 2020).

**PromptWizard**. A recent approach that jointly optimizes instructions and examples is PromptWizard (Agarwal et al., 2024). This is an important advancement as automatic prompt optimization

also covers optimization of the few-shot examples ("exemplar optimization"), i.e., improving the selection of relevant examples. Recent research by Wan et al. (2024) indicates that (1) even simple random example selection can yield performance improvements compared to sophisticated instruction optimization methods, and (2) combining instruction and example optimization creates synergistic effects enhancing overall performance.

The PromptWizard algorithm iteratively improves prompts through multiple steps: generating instruction variants via different thinking styles (mutation), evaluating them (scoring), providing feedback on top performers (critique), and implementing refinements (synthesis). It simultaneously optimizes in-context examples and uses critique and synthesis to produce synthetic examples addressing the prompt's weaknesses. Moreover, PromptWizard incorporates automatically generated chain-of-thought reasoning for few-shot examples and leverages task intent and an expert persona in prompts. It reportedly outperforms INSTINCT (Lin et al., 2024), InstructZero (Chen et al., 2024), APE (Zhou et al., 2023), PromptBreeder (Fernando et al., 2024), and EvoPrompt (Guo et al., 2024) on BIG-Bench Instruction Induction (BBII) while substantially reducing LLM calls and token usage.

## A.2 AutoML Techniques: Racing and Multi-Objective Optimization

As pointed out in the main paper, the field of AutoML offers many techniques that aim to make optimization more efficient. This includes racing algorithms (Maron and Moore, 1994; Birattari et al., 2002; López-Ibáñez et al., 2016), multi-fidelity optimization (Jamieson and Talwalkar, 2016; Li et al., 2018; Falkner et al., 2018; Awad et al., 2021), and multi-objective optimization with efficiency as an additional goal (Karl et al., 2023), to name just a few. These methods have also been successfully adopted beyond AutoML, for example, in the field of prompt selection, where efficiency is similarly important (Schneider et al., 2024; Shi et al., 2024). In the following, we provide more details on racing and multi-objective optimization, two AutoML techniques which we transfer to the field of prompt optimization with CAPO.

**Racing**. Racing refers to class of algorithms initially proposed for model selection in Machine Learning (Maron and Moore, 1994) and later adopted for algorithm configuration (Birattari et al., 2002). These algorithms sequentially evaluate candidates and eliminate poor one as soon as enough statistical evidence is collected against them, continuing the race only with surviving candidates. This approach accelerates optimization by spending fewer evaluations on poor candidates, allowing more resources to be concentrated on promising candidates (Birattari et al., 2002, 2010).

Hoeffding Races (Maron and Moore, 1994), one of the earliest racing methods, sequentially evaluate candidates on problem instances and use the Hoeffding's bound to eliminate statistically inferior options early. While this non-parametric approach imposes no distributional assumptions, it tends to be relatively conservative (Moore and Lee, 1994). BRACE (Moore and Lee, 1994) therefore uses Bayesian statistics instead of loose non-parametric bounds like Hoeffding's, enabling much earlier elimination of poor candidates.

F-Race (Birattari et al., 2002), forming the basis for many contemporary racing algorithms, employs the Friedman two-way analysis of variance by ranks (Conover, 1999), an omnibus test to compare multiple candidates across multiple problem instances. It ranks the candidates' performances within each instance to build cumulative evidence of which configurations are superior. It tests against the null hypothesis that all possible candidate rankings are equally likely. If this hypothesis is rejected, pairwise post-hoc tests between individual candidates are performed with significantly worse candidates being eliminated. Otherwise, all candidates advance to the next step. Since F-Race is suitable only for moderate numbers of candidates, Iterative F-Race (I/F-Race) (Balaprakash et al., 2007) extends it by iteratively applying F-Race while updating a probabilistic model of the candidate space to assign more probability mass to promising regions, from which subsequent candidates are sampled.

The *irace* package (López-Ibáñez et al., 2016) provides a general iterated racing implementation, of which I/F-Race is a special case, and offers several extensions and improvements. It implements the paired t-test as an alternative to the Friedman test. The latter is preferable when score ranges across different instances are not commensurable or the objective is an order statistic, while the t-test is more suitable when the objective corresponds to the mean of the score function. For multi-class tasks, irace recommends structuring instances in blocks rather than adding single instances per iteration. At the end of a race, the surviving candidates with highest overall rank across all instances/blocks are selected. They also present elitist racing as extension, which protects high-performing candidates ("elites") from elimination unless a new candidate demonstrates superior performance across at least the same number of evaluation instances.

Lastly, we also present some approaches related to racing. FocusedILS, an instantiation of ParamILS (Hutter et al., 2009), employs an approach similar to racing to save evaluation costs by adaptively increasing the number of evaluations and comparing configurations based on domination: one configuration dominates another when it performs at least as well on the same number of instances. A "bonus run" mechanism allocates more evaluation resources to promising configurations. Similarly, Random Online Adaptive Racing (ROAR) and Sequential Model-based Algorithm Configuration (SMAC) (Hutter et al., 2011) implement an "intensification" mechanism. Although labeled as racing, these algorithms do not use statistical testing. If a new candidate performs worse than the incumbent on the set of common instances, evaluating the new candidate immediately stops. Otherwise, the mechanism adds further evaluations exponentially.

**Multi-Objective Optimization**. Multi-objective optimization addresses scenarios with multiple competing objectives. Typical applications involve balancing different prediction performance metrics or trading off predictive performance against computational efficiency, interpretability, or sparseness. Multi-objective approaches are commonly categorized into *a priori* and *a posteriori* methods (Karl et al., 2023).

A priori methods transform multiple objectives into a single one, for example, using a weighted sum of the objectives (scalarization), yielding only a single solution candidate (Karl et al., 2023). Although a single objective simplifies the optimization problem (Miettinen, 1998), this approach requires the weights to be chosen a priori, which can be non-trivial, and trade-offs between competing objectives cannot be fully captured by a single solution (Jin and Sendhoff, 2008).

Conversely, a posteriori methods produce a set of Pareto-optimal solutions that domain experts can analyze after the optimization process (Karl et al., 2023). Evolutionary algorithms are particularly well-suited due to their population-based nature. Notable multi-objective evolutionary optimizers include NSGA-II (Deb et al., 2002), which uses non-dominated sorting rank and crowding distance for selection, and SMS-EMOA (Beume et al., 2007), which employs marginal hypervolume contribution as secondary criterion. Bayesian Optimization approaches have also been extended to multi-objective scenarios, with ParEGO (Knowles, 2006) being a prominent example. ParEGO approximates the Pareto-front by utilizing a set of randomly generated scalarization weights throughout its iterations.

Finally, combinations of multi-objective optimization and racing methods have been developed. irace can be used to configure multi-objective optimization algorithms by converting multi-objective problems into single-objective evaluations using hypervolume or the $\varepsilon$-measure (López-Ibáñez et al., 2016). S-Race (Zhang et al., 2013), specifically designed for multiple objectives, discards candidates once there is sufficient statistical evidence against them with respect to all objectives, later extended by SPRINT-Race (Zhang et al., 2015a) and I/S-Race (Miranda et al., 2015). A multi-objective variant of ParamILS, MO-ParamILS (Blot et al., 2016), also exists, which works on a set of non-dominated configurations in the Pareto-sense ("archive") instead of a single configuration. MO-SMAC (Rook et al., 2025) extends the SMAC framework to multi-objective scenarios via a hypervolume-based acquisition function and a procedure for managing non-dominated configuration sets.

## B Algorithm Details

---

**Algorithm 2** CAPO Functions

---

**Require:** population $\mathcal{P}_\mu$, meta-LLM $\Phi_{\text{meta}}$, evaluation-LLM $\Phi_{\text{eval}}$, cross-over-meta-prompt $p_C$, mutation-meta-prompt $p_M$, number of crossovers $c$, offspring prompts $\mathcal{P}_{\text{off}}$, few-shot dataset $\mathcal{D}_{\text{shots}}$, maximum number of few-shot examples $k_{\max}$, blocks $\mathcal{B}$, confidence level $\alpha$, token length penalty control parameter $\gamma$, number of survivors $n_{\text{survive}}$, max. number of evaluated blocks $z_{\max}$

1: **function** CROSS_OVER($\mathcal{P}_\mu$, $\Phi_{\text{meta}}$, $p_C$, $c$)
2:     $\mathcal{P}_{\text{off}} \leftarrow []$
3:     **for** $j = 1$ to $c$ **do**
4:         $p_a, p_b \leftarrow$ SAMPLE($\mathcal{P}_\mu$, 2)         ▷ Sample without replacement; $p_a = (i_a, \boldsymbol{e_a})$, $p_b = (i_b, \boldsymbol{e_b})$
5:         $i_{\text{off}} \leftarrow \Phi_{\text{meta}}(p_C||i_a||i_b)$         ▷ Let meta-LLM cross the parent prompts
6:         $\boldsymbol{e}_{\text{off}} \leftarrow$ SAMPLE($\boldsymbol{e_a} \cup \boldsymbol{e_b}$, $\left\lfloor \frac{|\boldsymbol{e_a}|+|\boldsymbol{e_b}|}{2} \right\rfloor$)     ▷ Sample from parent shots without replacement
7:         $p_{\text{off}} \leftarrow (i_{\text{off}}, \boldsymbol{e}_{\text{off}})$
8:         $\mathcal{P}_{\text{off}} \leftarrow$ APPEND($p_{\text{off}}$, $\mathcal{P}_{\text{off}}$)
9:     **end for**
10:     **return** $\mathcal{P}_{\text{off}}$
11: **end function**
12: **function** MUTATE($\mathcal{P}_{\text{off}}$, $\Phi_{\text{meta}}$, $\Phi_{\text{eval}}$, $p_M$, $\mathcal{D}_{\text{shots}}$, $k_{\max}$)
13:     $\mathcal{P}_{\text{mut}} \leftarrow []$
14:     **for** $p_{\text{off}} \in \mathcal{P}_{\text{off}}$ **do**
15:         $i_{\text{mut}} \leftarrow \Phi_{\text{meta}}(p_M \, || \, i_{\text{off}})$         ▷ Let meta-LLM mutate the instruction
16:         $r \sim$ Unif($\{0, 1, 2\}$)
17:         **if** $r = 0$ and $|\boldsymbol{e}_{\text{off}}| < k_{\max}$ **then**         ▷ Case 1: Create a new few-shot example
18:             $\boldsymbol{e}_{\text{new}} \leftarrow \boldsymbol{e}_{\text{off}} \cup$ CREATE_SHOTS($\mathcal{D}_{\text{shots}}$, 1, $i_{\text{mut}}$, $\Phi_{\text{eval}}$)
19:         **else if** $r = 1$ and $|\boldsymbol{e}_{\text{off}}| > 0$ **then**         ▷ Case 2: Remove a few-shot example
20:             $\boldsymbol{e}_{\text{new}} \leftarrow$ SAMPLE($\boldsymbol{e}_{\text{off}}$, $|\boldsymbol{e}_{\text{off}}| - 1$)
21:         **end if**         ▷ Case 3: Keep number of few-shot examples
22:         $p_{\text{mut}} \leftarrow \big( i_{\text{mut}}, \text{SHUFFLE}(\boldsymbol{e}_{\text{new}}) \big)$
23:         $\mathcal{P}_{\text{mut}} \leftarrow$ APPEND($p_{\text{mut}}$, $\mathcal{P}_{\text{mut}}$)
24:     **end for**
25:     **return** $\mathcal{P}_{\text{mut}}$
26: **end function**
27: **function** DO_RACING($\mathcal{P}_\mu$, $\mathcal{B}$, $\Phi_{\text{eval}}$, $\alpha$, $\gamma$, $n_{\text{survive}}$, $z_{\max}$)
28:     $j \leftarrow 0$
29:     SHUFFLE($\mathcal{B}$)         ▷ Optional (hyperparameter)
30:     **while** $|\mathcal{P}_\mu| > n_{\text{survive}}$ and $j < z_{\max}$ **do**
31:         $j \leftarrow j + 1$
32:         $S \leftarrow$ EVALUATE($\mathcal{P}_\mu$, $B_{:j}$, length_penalty $= \gamma$)         ▷ Note: cache already evaluated blocks
33:         $\mathcal{P}_\mu \leftarrow$ RACING_ELIMINATION($\mathcal{P}_\mu$, $S$, $\alpha$, $n_{\text{survive}}$)
34:     **end while**
35:     $\mathcal{P}_\mu \leftarrow$ SORT($\mathcal{P}_\mu$)$[: n_{\text{survive}}]$         ▷ Make sure to return only $n_{\text{survive}}$ prompts
36:     **return** $\mathcal{P}_\mu$
37: **end function**
38: **function** RACING_ELIMINATION($\mathcal{P}_\mu$, $S$, $\alpha$, $n_{\text{survive}}$)
39:     $\mathcal{P}_{\text{survivors}} \leftarrow \mathcal{P}_\mu$
40:     $c_\alpha \leftarrow$ GET_CRITICAL_VALUE($\alpha$)
41:     **for** $p_i \in \mathcal{P}_{\text{survivors}}$ **do**
42:         $n_{\text{sig\_better}} \leftarrow \sum_{j \neq i} \mathbb{I}\{\text{GET\_TEST\_STATISTIC}(\boldsymbol{s_j}, \boldsymbol{s_i}) > c_\alpha\}$     ▷ Perform significance tests
43:         **if** $n_{\text{sig\_better}} \geq n_{\text{survive}}$ **then**
44:             $\mathcal{P}_{\text{survivors}} \leftarrow \mathcal{P}_{\text{survivors}} \setminus \{p_i\}$         ▷ Eliminate $p_i$
45:         **end if**
46:     **end for**
47:     **return** $\mathcal{P}_{\text{survivors}}$
48: **end function**

---

## C  Technical Details

### C.1  Model Details

We report detailed IDs and revisions of the utilized LLMs from HuggingFace in Table 4. To locally host the LLMs, we use vLLM 0.7.3 (Kwon et al., 2023) as fast and easy-to-use library for LLM inference and serving since it efficiently manages the required memory and allows the usage of quantized models. Note that we restrict maximum output length to 2048, which is long enough for almost all generations while still allowing for reasonable large batch sizes. The optimal batch size is chosen by vLLM depending on available memory.

Table 4: Overview of the utilized LLMs.

| Model | Huggingface ID | Revision |
|---|---|---|
| **Llama-3.3-70B** | shuyuej/Llama-3.3-70B-Instruct-GPTQ | 3a7f7f7d46e362291821aaefb0a38b632f1190a8 |
| **Qwen2.5-32B** | Qwen/Qwen2.5-32B-Instruct-GPTQ-Int4 | c83e67dfb2664f5039fd4cd99e206799e27dd800 |
| **Mistral-Small-24B** | ConfidentialMind/Mistral-Small-24B-Instruct-2501_GPTQ_G128_W4A16_MSE | 803393813b8fc4046fb663af2e3c56339a5b520b |

### C.2  Dataset Details

In our experiments we utilize five datasets, all retrieved from HuggingFace:

(1) SST-5 (Socher et al., 2013): sentiment classification dataset from the Stanford Sentiment Treebank (SST) with five different sentiment classes. The input $x$ is taken from the column "text", the labels $y$ from the column "label_text".

(2) AG News (Zhang et al., 2015b): topic classification dataset with titles and descriptions of news articles that are to be assigned to either *World, Sports, Business* or *Sci/Tech*. The input $x$ is taken from the column "text", the labels $y$ from the column "label_text".

(3) Subj (Pang and Lee, 2004): subjectivity classification dataset with movie reviews that are to be classified as either *subjective* or *objective*. The input $x$ is taken from the column "text", the labels $y$ from the column "label_text".

(4) GSM8K (Cobbe et al., 2021): grade school math word problems requiring multi-step reasoning. We utilize the train and test split of the "main" subset, from which the column "question" is used as input $x$, the label $y$ is extracted from the "answer" after ####.

(5) (Balanced) COPA (Kavumba et al., 2019): commonsense causal reasoning dataset with premises for which the plausible cause or effect is to be chosen from two alternatives. We create the input $x$ by concatenating the columns "premise", "question", "choice1", and "choice2" as follows: "<premise>\n <question> A: \n <choice1> \n <question> B: \n <choice2>". The labels $y$ are mapped from 0 and 1 in column "label" to "A" and "B".

We provide detailed IDs and revisions of the utilized datasets in Table 5. For $\mathcal{D}_{\text{shots}}$ and $\mathcal{D}_{\text{dev}}$, 500 instances are sampled from the train split without replacement with the random seed of the corresponding experiment. The first 300 points are used for $\mathcal{D}_{\text{dev}}$, the remaining 200 for $\mathcal{D}_{\text{shots}}$. To obtain $\mathcal{D}_{\text{test}}$, 500 instances are sampled from the test split and used throughout all experiments.

Table 5: Overview of the utilized HuggingFace datasets.

| Dataset | Huggingface ID | Revision | $n_{\text{train}}$ | $n_{\text{test}}$ | #classes |
|---|---|---|---|---|---|
| **SST-5** | SetFit/sst5 | e51bdcd8cd3a30da231-967c1a249ba59361279a3 | 8.5k | 2.2k | 5 |
| **AG News** | SetFit/ag_news | ca5ba619eb034211db5-f70932b6702efd21e7c73 | 120k | 7.6k | 4 |
| **Subj** | SetFit/subj | f3c1162e678417f664d-76b21864fdb87b0615fcf | 8k | 2k | 2 |
| **GSM8K** | openai/gsm8k | e53f048856ff4f594e95-9d75785d2c2d37b678ee | 7.5k | 1.3k | - |
| **COPA** | pkavumba/balanced-copa | 813bd03cd6e07d9bd8d7333896ad5d40abb95ea9 | 1k | 500 | 2 |

### C.3 Hardware Details

All computations are performed on a GPU cluster. For each experiment configuration, only a single GPU with at least 80GB of RAM (NVIDIA A100 (80GB) or NVIDIA H100 (94GB)) is used to host the corresponding LLM. Experiments are distributed across multiple instances for parallel execution. We report a total computation time of 13 GPU days for our experiments, not including the compute time for evaluation on hold-out test data.

### C.4 Implementation Details

**Answer Extraction**. To reliably extract information from LLM output in our experiments, we utilize marker-based extraction. Concretely, we parse the information in html-style tags: offspring/-mutated prompts are extracted between <prompt></prompt> markers and predictions between <final_answer></final_answer> markers in the LLM output. This information is also included in the initial instructions and task descriptions. Details and examples are provided in the subsequent sections of this appendix.

**Optimizer Parametrization**. For our experiments, we use the following default hyperparameters: We parametrize our CAPO algorithm with $\alpha = 0.2$, $b = 30$ and $z_{max} = 10$ (i.e., $b \cdot z_{max} = |\mathcal{D}_{dev}|$), $k_{max} = 5$, $\mu = 10$, $c = 4$, $\gamma = 0.05$ (a prompt with same length as the longest initial prompt (instruction + examples) is penalized by 5%p). Further, we use our simplified meta-prompts $p_C$ and $p_M$ (cf. Appendix D.3), a paired t-test for racing, and no block shuffling for cost-efficiency.

For EvoPromptGA (Guo et al., 2024), we also use a population size 10 following the recommendations of the original paper. For OPRO (Yang et al., 2024), also following the publication, we limit the number of previous prompts in the meta-prompt to 20, generate 8 new prompts per iteration, and use 3 few-shot examples in the meta-prompt. For PromptWizard (Agarwal et al., 2024), we use the original parametrization, and provide one randomly sampled instruction from our pool, our task description, and answer format. It is important to note that we cannot trivially extend PromptWizard to make use of the full budget in our experiments. In its original multi-step implementation, each algorithmic step (cf. Appendix A.1) is performed a specified number of iterations before continuing to the next one. Having a predefined maximum budget, it is unclear how to distribute it between the steps in advance.

**Optimizer Implementation**. For EvoPromptGA and OPRO, we use reimplementations that are available as part of a public library.[8] We note that this library is developed and maintained by the authors, allowing to directly ensure the correctness of the implementations. For PromptWizard, we utilize the original implementation[9] with small adaptions for our LLMs.

**Seeding**. For statistical robustness, we conduct three independent runs of each optimizer-LLM-dataset configuration with varying random seeds to quantify variance. Seeds influence stochastic elements of the optimizers, initial instruction selection, dev set sampling and LLM decoding.

**Budget and Prompt Length Computation**. We compute input token budget usage by applying each LLM's respective tokenizer and count the resulting number of tokens. Similarly, the prompt length penalty is also computed based on the number of tokens produced by the respective tokenizer. In contrast, to compute the prompt lengths reported in our results (Section 6 and Appendix F to I), we count the number of words in a prompt separated by whitespace to ensure comparability between LLMs.

---

[8] `https://github.com/finitearth/promptolution` (accessed: 2025-03-22)
[9] `https://github.com/microsoft/PromptWizard` (accessed: 2025-03-22)

# D Input Specifications and Templates

## D.1 Task Descriptions

Table 6: Manually created task descriptions used for CAPO, OPRO, and PromptWizard.

**SST-5:**
The dataset consists of movie reviews with five levels of sentiment labels: very negative, negative, neutral, positive, and very positive. The task is to classify each movie review into one of these five sentiment categories. The class will be extracted between the markers `<final_answer>answer/final_answer>`.

**AG News:**
The dataset contains news articles categorized into four classes: World, Sports, Business, and Sci/Tech. The task is to classify each news article into one of the four categories. The class will be extracted between the markers `<final_answer>answer</final_answer>`.

**Subj:**
The dataset contains sentences labeled as either subjective or objective. The task is to classify each sentence as either subjective or objective. The class will be extracted between the markers `<final_answer>answer</final_answer>`.

**GSM8K:**
The dataset consists of grade school math word problems that require multi-step reasoning to solve. The task is to solve each word problem and provide the final answer. The final solution will be extracted between the markers `<final_answer>answer</final_answer>`.

**(Balanced) COPA:**
The dataset consists of premises and two possible choices for the effect or cause of the premise. The task is to determine which of the two choices (A or B) is the correct effect of the premise. The class will be extracted between the markers `<final_answer>answer</final_answer>`.

## D.2 Initial Instructions

Since both CAPO and EvoPrompt require initial instructions to start from, we create a set of 15 initial instructions for each task. To demonstrate that this requirement of initial instructions is not a major limiting factor of the algorithms, we produce them in an automated manner, prompting Anthropic's Claude Sonnet 3.7 (`https://claude.ai/`) to create a diverse set of initial instructions, making use of our task descriptions in Appendix D.1. The full prompt template is provided in Table 7. Alternatively, approaches like APE (Zhou et al., 2023) could be employed to generate initial instructions, or they could be manually engineered, e.g., by domain experts, to incorporate specific prior knowledge. Examples of our initial instructions with corresponding test scores are given in Appendix J.1.

Table 7: Prompt used to generate initial instructions with Anthropic's Claude Sonnet 3.7. The `<task_description>` placeholder is replaced with our task description.

Please create diverse prompts for the following task. They should be linguistically diverse (but always in English) and have varying lengths and complexities. This means some consist only of a short sentence with a rather high-level description while others elaborate on the task in little more detail.
Task: `<task_description>`
Explicitly state this expected format as part of the prompts. Create overall 15 prompts within quotes as an array:

To generate generic, task-unspecific instructions for ablation study V. in Section 6.2, we use the "task description" in Table 8.

Table 8: Task Description for generation of "generic" initial instructions.

Create prompts that are so generic, they could work for almost any task. The answers provided by the LLM should be contained within `<final_answer> </final_answer>`.

## D.3 Meta-Prompt Templates

Table 9: List of all meta-prompt templates used in CAPO and EvoPromptGA. The purple text indicates placeholders where the according elements are inserted.

---

**CAPO cross-over meta-prompt template**:
You receive two prompts for the following task: `<task_description>`
Please merge the two prompts into a single coherent prompt. Maintain the key linguistic features from both original prompts:
Prompt 1: `<mother>`
Prompt 2: `<father>`

Return the new prompt in the following format:
`<prompt>new prompt</prompt>`.

---

**CAPO mutation meta-prompt template**:
You receive a prompt for the following task: `<task_description>`
Please rephrase the prompt, preserving its core meaning while substantially varying the linguistic style.
Prompt: `<instruction>`

Return the new prompt in the following format:
`<prompt>new prompt </prompt>`

---

**Original EvoPromptGA meta-prompt template from Guo et al. (2024)**:
Please follow the instruction step-by-step to generate a better prompt.
1. Crossover the following prompts and generate a new prompt:
Prompt 1: Rewrite the input text into simpler text.
Prompt 2: Rewrite my complex sentence in simpler terms, but keep the meaning.
2. Mutate the prompt generated in Step 1 and generate a final prompt bracketed with `<prompt>` and `<prompt>`.

1. Crossover Prompt: Rewrite the complex text into simpler text while keeping its meaning.
2. `<prompt>`Transform the provided text into simpler language, maintaining its essence.`<prompt>`

Please follow the instruction step-by-step to generate a better prompt.
1. Crossover the following prompts and generate a new prompt:
Prompt 1: `<prompt1>`
Prompt 2: `<prompt2>`
2. Mutate the prompt generated in Step 1 and generate a final prompt bracketed with `<prompt>` and `<prompt>`.

1.

---

**EvoPromptGA simplified meta-prompt template used in the ablation study in Appendix I.3**:
You receive two prompts for the following task: `<task_description>`
1. Please merge the two prompts into a single coherent prompt. Maintain the key linguistic features from both original prompts:
Prompt 1: `<prompt1>`
Prompt 2: `<prompt2>`

2. Please rephrase the prompt generated in step 1, preserving its core meaning while substantially varying the linguistic style.
Return the final prompt in the following format:
`<prompt>final prompt<prompt>`

---

CAPO performs cross-over and mutation separately, each with its own template, while EvoPromptGA (Guo et al., 2024) executes both operations with a single meta-prompt. We emphasize that the CAPO prompts are simplified and substantially shorter, i.e., need fewer input tokens, and they do not require any notion of what is a "good" prompt by avoiding terms like "better". In an additional experiment described in Appendix I.3, we use our simplified CAPO templates within EvoPromptGA. For this purpose, we combine them into a single meta-prompt also shown in Table 9.

# E Examples of CAPO Algorithm Operations

## E.1 Cross-over and Mutation Examples

Table 10: Concrete examples for cross-over and mutation with Mistral-Small-24B on COPA using CAPO. The purple text indicates the filled placeholders from the templates, green text marks the resulting response of the LLM.

---

**Crossover:**

You receive two prompts for the following task: The dataset consists of premises and two possible choices for the effect or cause of the premise. The task is to determine which of the two choices (A or B) is the correct effect of the premise. The class will be extracted between the markers <final_answer>answer</final_answer>.

Please merge the two prompts into a single coherent prompt. Maintain the key linguistic features from both original prompts:

Prompt 1: Select the statement that represents the most reasonable causal relationship to the given context. Respond with <final_answer>A</final_answer> or <final_answer>B</final_answer> only.

Prompt 2: Based on causal reasoning, which is more plausible: A or B? Enclose your answer with <final_answer> tags like this: <final_answer>A</final_answer> or <final_answer>B</final_answer>.

Return the new prompt in the following format:
<prompt>new prompt</prompt>

<prompt>Based on causal reasoning, select the statement that represents the most reasonable causal relationship to the given context. Which is more plausible: A or B? Enclose your answer with <final_answer> tags like this: <final_answer>A</final_answer> or <final_answer>B</final_answer>.</prompt>

---

**Mutation:**

You receive a prompt for the following task: The dataset consists of premises and two possible choices for the effect or cause of the premise. The task is to determine which of the two choices (A or B) is the correct effect of the premise. The class will be extracted between the markers <final_answer>answer</final_answer>.

Please rephrase the prompt, preserving its core meaning while substantially varying the linguistic style.

Prompt: Select the statement that represents the most reasonable causal relationship to the given context. Respond with <final_answer>A</final_answer> or <final_answer>B</final_answer> only.

Return the new prompt in the following format:
<prompt>new prompt</prompt>

<prompt>Identify the statement that best aligns with the cause of the given context. Provide your response as <final_answer>A</final_answer> or <final_answer>B</final_answer> only.</prompt>

---

## E.2 Exemplary Few-Shot Examples With and Without Reasoning

Table 11: Concrete examples for few-shot examples in CAPO with and without reasoning using Llama-3.3-70B on GSM8K and Subj.

---

**GSM8K - Few-shot example with reasoning**:

Input: In 10 years, Terry will be 4 times the age that Nora is currently. If Nora is currently 10 years old, how old is Terry now?

Output: To solve this problem, let's break it down step by step.

1. We know that Nora is currently 10 years old.

2. In 10 years, Terry will be 4 times the age that Nora is currently. Since Nora is currently 10 years old, 4 times her current age is 4 * 10 = 40.

3. This means that in 10 years, Terry will be 40 years old.

4. To find Terry's current age, we need to subtract 10 years from the age Terry will be in 10 years. So, Terry's current age is 40 - 10 = 30.

Therefore, `<final_answer>30</final_answer>` is Terry's current age.

---

**GSM8K - Few-shot example without reasoning**:

Input: Kendra has 4 packs of pens. Tony has 2 packs of pens. There are 3 pens in each pack. If Kendra and Tony decide to keep two pens each and give the remaining pens to their friends one pen per friend, how many friends will they give pens to?

Output: `<final_answer>14</final_answer>`

---

**Subj - Few-shot example with reasoning**:

Input: gangs , despite the gravity of its subject matter , is often as fun to watch as a good spaghetti western .

Output: The given sentence is subjective because it expresses a personal opinion by comparing the experience of watching "gangs" to a "good spaghetti western" and describing it as "fun to watch." This comparison and the use of the word "fun" introduce a personal judgment about the entertainment value of the subject matter, which may vary from person to person.

`<final_answer>` Subjective `</final_answer>`

---

**Subj - Few-shot example without reasoning**:

Input: this holds particularly true for blacky , a white teen who is more interested in books than sport , and his best friend , dumby , the aboriginal star of the team .

Output: `<final_answer>objective</final_answer>`

---

## F Hyperparameter Sensitivity Analysis

In this section, we investigate the univariate effects of hyperparameters in CAPO. The hyperparameters we alter are the length penalty $\gamma$ (0.0, 0.01, 0.02, 0.05, 0.1), significance level $\alpha$ (0.05, 0.1, 0.2, 0.5), population size $\mu$ (6, 8, 10, 12), cross-overs per iteration $c$ (4, 7, 10), and whether we shuffle the blocks in racing or not. In each case, we hold all other hyperparameters fixed to their defaults (cf. Appendix C.4). Thus, multivariate dependencies are not considered here. All experiments are conducted with Llama-3.3-70B model on two datasets (AG News and GSM8K). The budget is limited to 5M input tokens, and each configuration is executed with three different seeds. We summarize our results in Table 12.

The results indicate that our default parameters are not optimal for neither AG News nor GSM8K as they are outperformed by other parametrizations. However, performance differences for all parameter variations lie within one standard deviation. We conclude that while hyperparameters influence the final performance, their impact is rather moderate. Since changing individual parameters affects not only the final performance but also the behavior of the optimization process, we provide test score curves below.

Table 12: Hyperparameter sensitivity analysis of various CAPO parametrizations with Llama-3.3-70B after 5M input tokens. We report mean accuracy (in %) with standard deviations on test set for the best prompts across three seeds. The best prompt per seed is selected from the final population based on the available development set scores. Hyperparameters are varied univariately, keeping all other parameters at their defaults. Bold values indicate best performance for each parameter and task.

| Parametrization | AG News | GSM8K | Avg. |
|---|---|---|---|
| $\gamma$=0 | 89.27±0.41 | 74.93±1.04 | 82.10 |
| $\gamma$=0.01 | **89.53**±0.25 | **75.27**±3.10 | **82.40** |
| $\gamma$=0.02 | 89.20±0.43 | 74.20±3.28 | 81.70 |
| $\gamma$=0.05 *(default)* | 88.80±0.75 | 73.37±3.73 | 81.27 |
| $\gamma$=0.1 | 88.73±1.11 | 74.80±3.15 | 81.77 |
| $\alpha$=0.05 | **89.20**±0.59 | 73.87±2.17 | 81.53 |
| $\alpha$=0.1 | 88.93±0.62 | 74.87±2.79 | **83.00** |
| $\alpha$=0.2 *(default)* | 88.80±0.75 | 73.73±3.73 | 81.90 |
| $\alpha$=0.5 | 87.40±2.37 | **75.93**±1.51 | 81.67 |
| $\mu$=6 | 89.00±0.49 | **77.67**±3.03 | **83.33** |
| $\mu$=8 | 88.33±0.25 | **77.67**±3.74 | 83.00 |
| $\mu$=10 *(default)* | 88.80±0.75 | 73.73±3.73 | 81.27 |
| $\mu$=12 | **89.33**±0.19 | 76.87±1.31 | 83.10 |
| $c$=4 *(default)* | 88.80±0.75 | 73.73±3.73 | 81.27 |
| $c$=7 | 89.47±0.25 | 73.07±1.64 | 81.27 |
| $c$=10 | **89.53**±0.19 | **74.40**±3.30 | **81.97** |
| w/ shuffling | **89.60**±0.28 | **76.73**±1.81 | **83.17** |
| w/o *(default)* | 88.80±0.75 | 73.73±3.73 | 81.27 |

A smaller length penalty $\gamma$ naturally improves performance (cf. Figure 5) since the prompt length becomes less influential to the optimization process allowing for longer, often better performing prompts. Figure 6 shows that for larger length penalties, prompt lengths decrease as optimization advances before stabilizing, which aligns with expected behavior. However, a trade-off exists since long prompts consume significant portions of the budget and therefore permit fewer steps within the same budget constraints.

The choice of the significance level $\alpha$ used in the paired t-test for racing governs how strictly underperforming prompts are eliminated: lower values lead to more conservative eliminations while higher values allow for more aggressive pruning. As shown in Table 12, $\alpha = 0.1$ yields

the highest average performance across tasks with our default setting of $\alpha = 0.2$ still within one standard deviation. Notably, $\alpha = 0.05$ performs best on AG News, whereas $\alpha = 0.5$ achieves the top result on GSM8K. Nonetheless, while $\alpha$ does influence optimization dynamics, especially the trade-off between exploration and premature elimination, its overall effect on final performance remains moderate. The optimization curves in Figure 7 illustrate the effect of $\alpha$: while larger values allow for more steps as prompts are eliminated early, the corresponding optimization curves increase more slowly (for GSM8K) or not all (for AG News) as the probability of falsely eliminating a good prompt is considerably higher.

Choosing the optimal population size $\mu$ depends on the task. Large $\mu$ improves performance on AG News while a small $\mu$ is beneficial on GSM8K. Looking at Table 12, we observe that this hyperparameter choice has the largest impact on the average test set performance of the best candidates per seed. The smaller the population size, the more steps can be performed, which is again a trade-off. For small population sizes, there is a danger of getting "stuck" when there is insufficient diversity in the prompts to create new explorative candidates. We can see this effect in Figure 8 for AG News at $\mu = 6$. We also observe a larger standard deviation for smaller population sizes.

The number of cross-overs per iteration has a minor influence on final performance. On our two datasets, we observe slight improvements for larger $c$. In general, for smaller $c$, more steps are possible and standard deviations are smaller (cf. Figure 9). An important consideration is that with large $c$, promising prompts from previous populations are more likely to be erroneously eliminated in racing despite being superior, as it may be eliminated on early blocks.

Shuffling the blocks during racing slightly improves the performance on both tasks. A potential explanation is that shuffling prevents overfitting to early blocks. However, this approach has the drawback that fewer steps are possible (cf. Figure 10) since we cannot always use cached evaluations and therefore cannot perform as many steps as without shuffling.

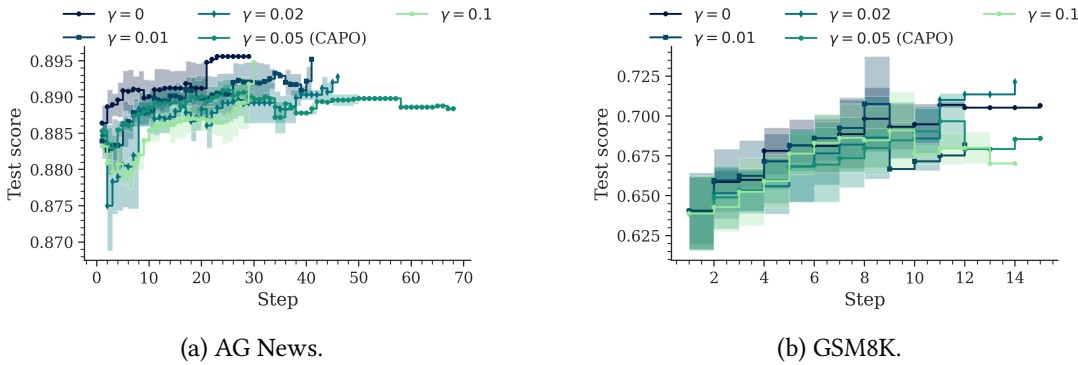

(a) AG News.  (b) GSM8K.

Figure 5: Population mean test scores over steps with Llama-3.3-70B. Mean and standard deviations are computed across seeds. We univariately vary the length penalty $\gamma$ keeping all other parameters at their defaults.

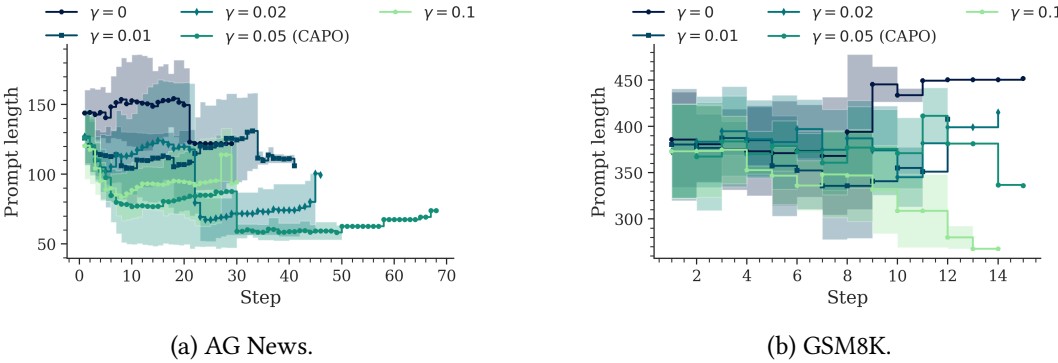

(a) AG News.

(b) GSM8K.

Figure 6: Population mean prompt lengths over steps with Llama-3.3-70B. Mean and standard deviations are computed across seeds. We univariately vary the length penalty $\gamma$ keeping all other parameters at their defaults.

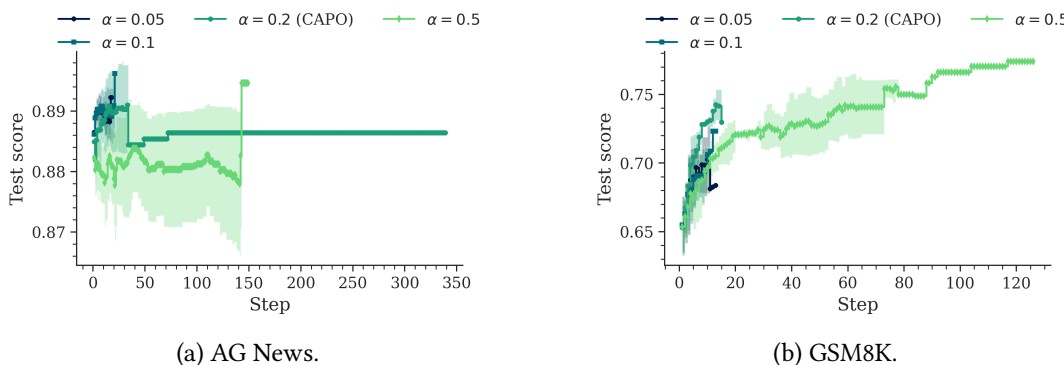

(a) AG News.

(b) GSM8K.

Figure 7: Population mean test scores over steps with Llama-3.3-70B. Mean and standard deviations are computed across seeds. We univariately vary the significance level $\alpha$ keeping all other parameters at their defaults.

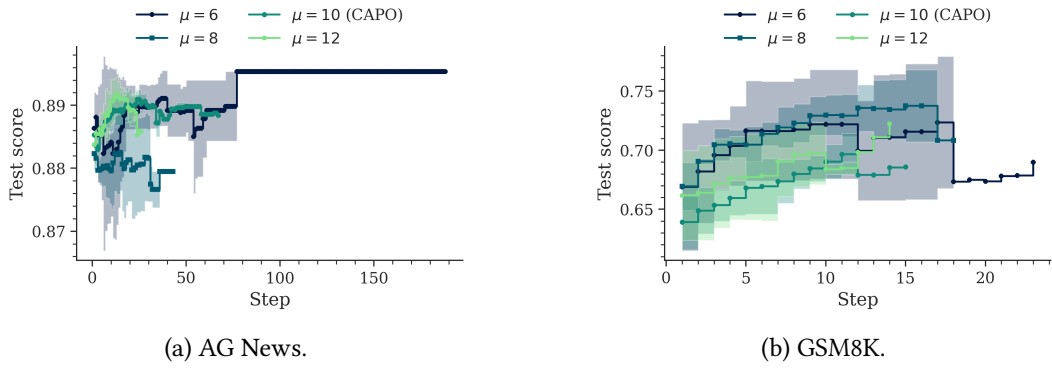

(a) AG News.

(b) GSM8K.

Figure 8: Population mean test scores over steps with Llama-3.3-70B. Mean and standard deviations are computed across seeds. We univariately vary the population size $\mu$ keeping all other parameters at their defaults.

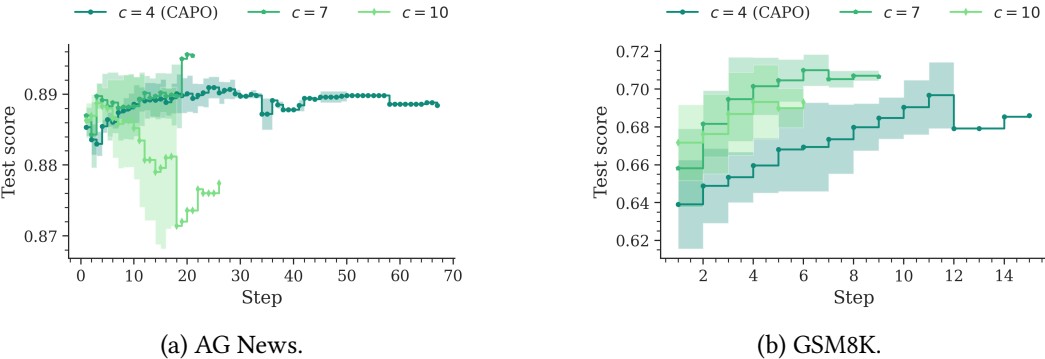

(a) AG News.

(b) GSM8K.

Figure 9: Population mean test scores over steps with Llama-3.3-70B. Mean and standard deviations are computed across seeds. We univariately vary the number of crossovers $c$ keeping all other parameters at their defaults.

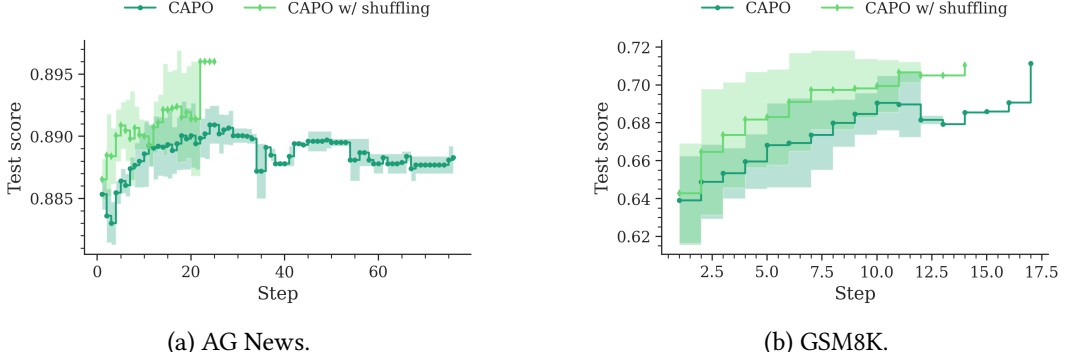

(a) AG News.

(b) GSM8K.

Figure 10: Population mean test scores over steps with Llama-3.3-70B. Mean and standard deviations are computed across seeds. We compare CAPO with vs. without (default) shuffling of the blocks during racing CAPO.

## G CAPO Detailed Analysis

### G.1 Token Usage Breakdown of Evaluation-LLM vs. Meta-LLM

In the following, we analyze the proportion of input tokens consumed by the evaluation-LLM compared to the meta-LLM across various tasks (cf. Table 13). On average, the evaluation-LLM accounts for 96.6% of the total token usage. For most datasets, this proportion exceeds 98%. The only notable exception is COPA, for which the share of the evaluation-LLM drops slightly for some models. This consistently high share justifies our approach of mainly considering the evaluation-LLM in our cost considerations.

Table 13: Proportion of input tokens consumed by the evaluation-LLM compared to the meta-LLM across the benchmark experiments in Section 6.1.

| Model | AG News | COPA | GSM8K | SST-5 | Subj |
|---|---|---|---|---|---|
| **Llama** | 97.9 % | 98.4% | 99.6% | 98.9% | 98.6% |
| **Mistral** | 98.8% | 73.4% | 99.7% | 99.1% | 99.0% |
| **Qwen** | 99.0% | 89.5% | 99.6% | 99.1% | 98.8% |

### G.2 Influence of Few-Shot Examples on Prompt Length

We find that CAPO allocates prompt length adaptively across tasks, with few-shot examples contributing substantially more to the total prompt length on complex tasks. For instance, on GSM8K, over 80% of the final prompt length stems from few-shot examples, compared to less than 52% on COPA. Averaged across all datasets, few-shot examples account for 66% of the total prompt length.

Table 14: Instruction length, few-shot length, and percentage of few-shot content of the best prompts generated by CAPO across different tasks. Mean and standard deviation are computed across three seeds. The best prompt per seed is selected from the final population based on development set scores. The system prompts are counted as part of the instructions.

| Model | Task | Instruction | Few-shots | % Few-shots |
|---|---|---|---|---|
| Llama-3.3-70B | SST-5 | 52±24 | 109± 61 | 68± 3 |
| | AG News | 76±41 | 34± 25 | 31±25 |
| | Subj | 56±27 | 102± 22 | 65±16 |
| | GSM8K | 37±14 | 444±126 | 92± 1 |
| | COPA | 40±21 | 43± 29 | 51±23 |
| Qwen2.5-32B | SST-5 | 74± 7 | 114± 21 | 61±15 |
| | AG News | 63±30 | 53± 28 | 46±28 |
| | Subj | 86±14 | 72± 13 | 46± 8 |
| | GSM8K | 40± 8 | 190± 98 | 83±18 |
| | COPA | 97±47 | 8± 10 | 8± 7 |
| Mistral-Small-24B | SST-5 | 56± 1 | 86± 19 | 61±20 |
| | AG News | 56±19 | 98± 69 | 64±35 |
| | Subj | 35± 8 | 103± 35 | 75± 5 |
| | GSM8K | 38± 9 | 247± 18 | 87± 2 |
| | COPA | 59± 3 | 18± 24 | 23±20 |

### G.3 Prompt Survival Analysis

Figure 11 shows how the population evolves over multiple steps for two examples with different models and datasets. The visualization tracks test performance for all population members, distinguishing between surviving prompts, newly proposed candidates, and eliminated (killed) prompts in each step.

In the early optimization phases, we observe the generation of relatively low-performing prompts, which the algorithm correctly eliminates. As optimization progresses, the quality of newly proposed prompts gradually improves. Since the algorithm does its selection based on the development set scores it can happen that a prompt, which would have performed better on the test set, gets eliminated (cf. Figure 11a).

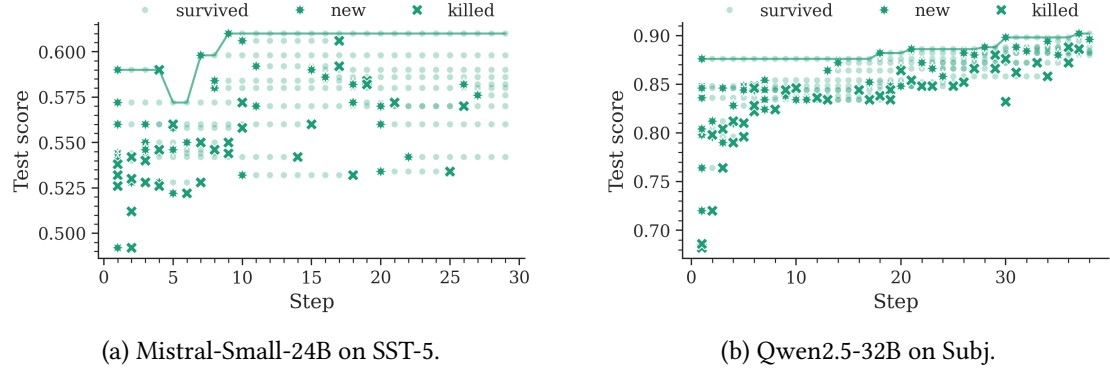

(a) Mistral-Small-24B on SST-5.  (b) Qwen2.5-32B on Subj.

Figure 11: Test scores of all population members over steps of default CAPO for one seed (42). Every time a prompt is newly proposed or gets killed this is indicated by a special marker. The line at the upper end shows the progression of the current best prompt.

# H  Further Benchmark Results

## H.1  Performance Profile

The performance profile plot displays the frequency $\rho(\tau)$ of an optimization algorithm producing an instance with a performance difference of $\tau$ to the best performing instance. For each dataset-model pair, we compute the average performance across seeds, using the best-performing prompts selected from the final optimization step on the dev-set. Each of these averaged results serves as an instance in our analysis. While the original proposal introduced by Dolan and Moré (2002) uses the ratio to the maximum performance, we follow Agarwal et al. (2024) and Lin et al. (2024) and report the difference to the best performing prompt, as the accuracy metric is bounded between 0 and 1.

Thus we get for distance $\tau$, optimizer $\Psi$, performance on task $i$ with optimizer $\psi$ $\sigma_{i,\psi}$ and number of tasks $n$:

$$\rho_\Psi(\tau) = \frac{1}{n} \sum_{i=1}^{n} \mathbb{I}[\sigma_{i,\max} - \sigma_{i,\Psi} \leq \tau]. \tag{3}$$

Therefore, $\rho_\Psi(0)$ indicates the frequency of optimizer $\Psi$ producing the best instance per task. Figure 12 shows, that with a $\rho_{\text{CAPO}}(0.012) = 1$ we are within 1.2 %p of the best performing instance in every single task-model pair.

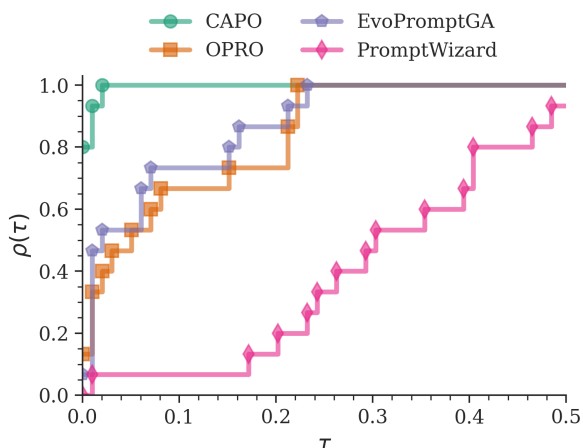

Figure 12: Performance profiles of all benchmarked optimizers.

## H.2 Further Optimization Curves from Benchmark Experiments

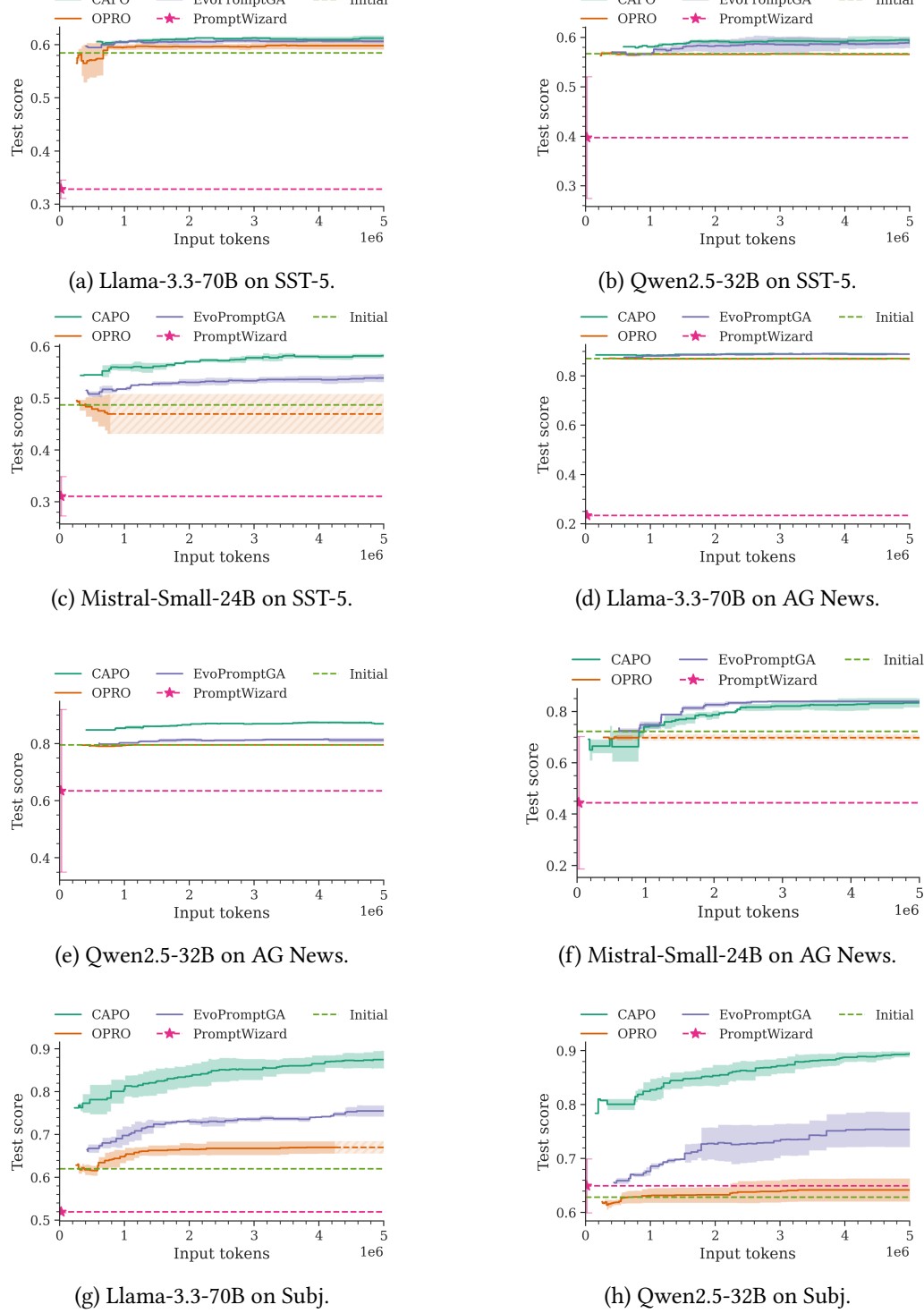

(a) Llama-3.3-70B on SST-5.

(b) Qwen2.5-32B on SST-5.

(c) Mistral-Small-24B on SST-5.

(d) Llama-3.3-70B on AG News.

(e) Qwen2.5-32B on AG News.

(f) Mistral-Small-24B on AG News.

(g) Llama-3.3-70B on Subj.

(h) Qwen2.5-32B on Subj.

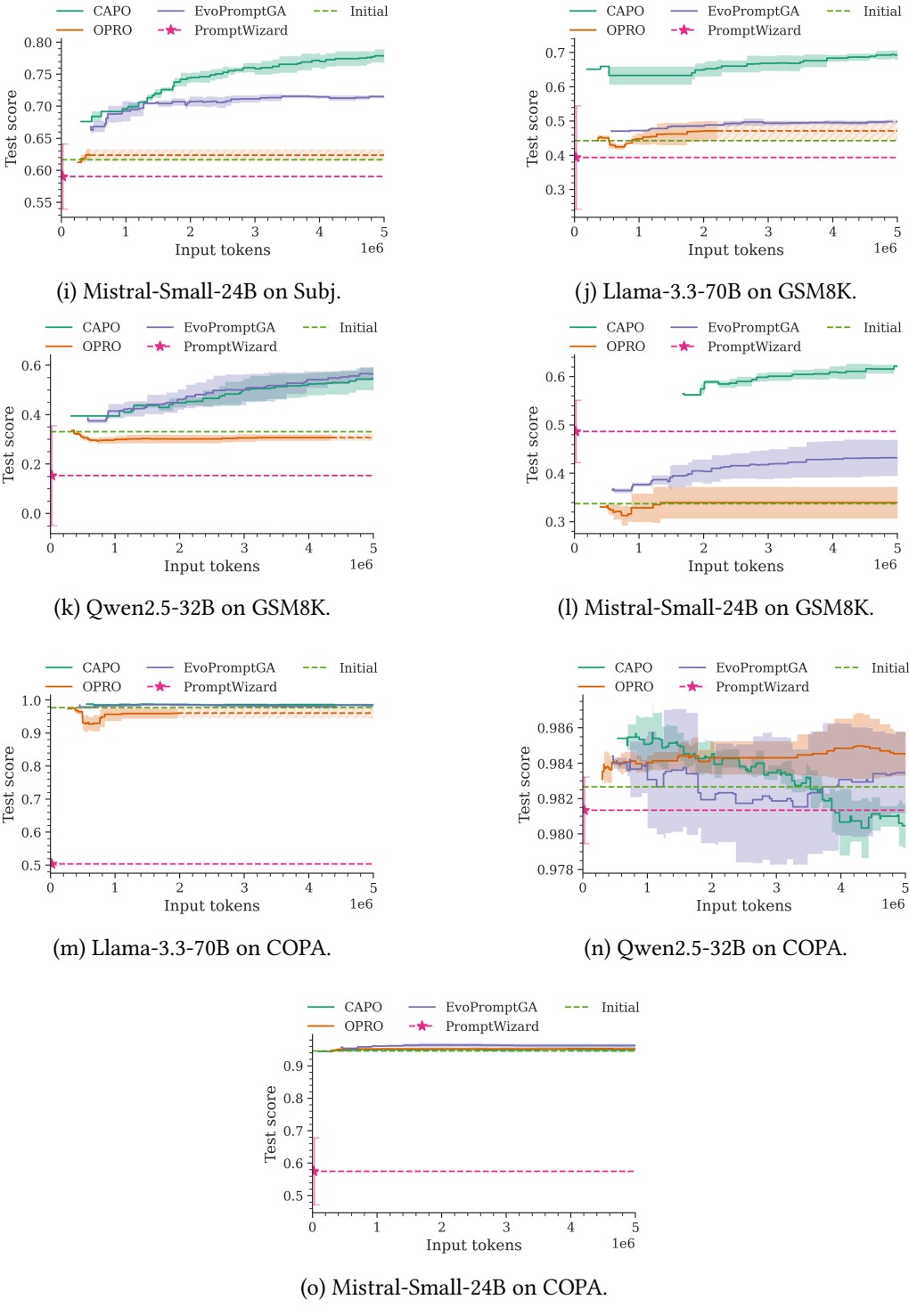

(i) Mistral-Small-24B on Subj.

(j) Llama-3.3-70B on GSM8K.

(k) Qwen2.5-32B on GSM8K.

(l) Mistral-Small-24B on GSM8K.

(m) Llama-3.3-70B on COPA.

(n) Qwen2.5-32B on COPA.

(o) Mistral-Small-24B on COPA.

Figure 13: Population mean test scores over input tokens from benchmark experiments for all datasets and models. Mean and standard deviations are computed across seeds. PromptWizard produces prompts only once after a small number of input tokens, marked with a star (mean) and error bars (std). If an algorithm converges (i.e., when it outputs the same prompts each step, which can happen for OPRO), we continue the curve with a dashed horizontal line and hatched area.

## H.3 Prompt Lengths from Benchmark Experiments

Table 15: Mean prompt length with standard deviation of the best prompts for different optimization methods, datasets, and models. Mean and standard deviation are computed across three seeds. The best prompt per seed is selected from the final population based on the available development set scores (for CAPO: penalized average block scores of evaluated blocks). Bold values indicate shortest prompts. The system prompts are counted as part of the prompt.

| Model | Optimizer | SST-5 | AG News | Subj | GSM8K | COPA | Avg. |
|---|---|---|---|---|---|---|---|
| **Llama-3.3-70B** | Initial | 33± 5 | 35± 6 | 31± 8 | 29± 7 | **30**± 5 | 32 |
| | OPRO | 63± 22 | 32± 4 | 42± 4 | 58± 15 | 33± 7 | 46 |
| | PromptWizard | 563± 36 | 1106±265 | 863±400 | 544±173 | 613± 33 | 738 |
| | EvoPromptGA | **33**± 2 | **30**± 1 | **28**± 2 | **28**± 2 | 32± 2 | **29** |
| | CAPO (ours) | 161± 85 | 110± 46 | 158± 12 | 481±113 | 83± 22 | 199 |
| **Qwen2.5-32B** | Initial | 33± 5 | **35**± 6 | **31**± 8 | 29± 7 | **30**± 5 | **32** |
| | OPRO | 38± 5 | 37± 8 | 33± 5 | 27± 2 | 51± 14 | 37 |
| | PromptWizard | 677±517 | 753±541 | 297± 22 | 698±392 | 337± 32 | 552 |
| | EvoPromptGA | **37**± 4 | **35**± 6 | 35± 5 | **25**± 6 | 40± 9 | 34 |
| | CAPO (ours) | 187± 28 | 116± 56 | 158± 13 | 230± 89 | 105± 49 | 159 |
| **Mistral-Small-24B** | Initial | 33± 5 | **35**± 6 | 31± 8 | 29± 7 | **30**± 5 | 32 |
| | OPRO | **29**± 2 | 44± 7 | 26± 0 | 32± 10 | 36± 5 | 33 |
| | PromptWizard | 1027±246 | 544±214 | 701±297 | 579±112 | 1139±188 | 798 |
| | EvoPromptGA | **29**± 2 | 39± 9 | **26**± 1 | **20**± 1 | 31± 2 | **29** |
| | CAPO (ours) | 142± 21 | 153± 78 | 138± 39 | 286± 24 | 76± 27 | 159 |

# I Further Ablation Results

## I.1 Optimization Curves from Ablation Studies

For all plots of the mean test scores over input tokens, the mean and standard deviations are computed across seeds. If an algorithm run terminates early, we continue the curve with a dashed horizontal line and hatched area.

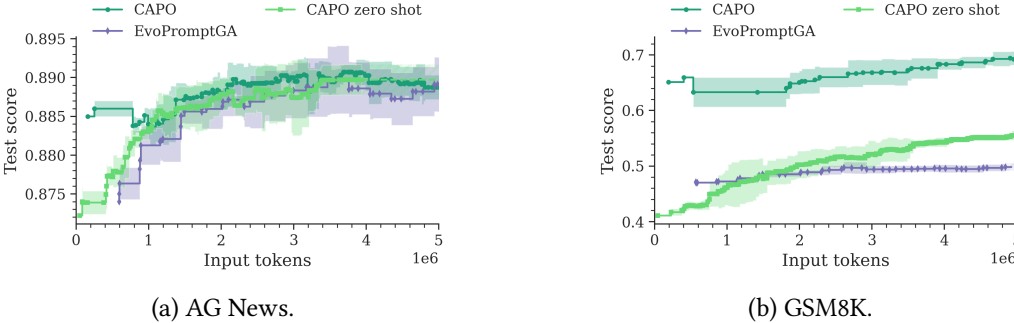

(a) AG News.

(b) GSM8K.

Figure 14: Population mean test scores over input tokens with Llama-3.3-70B. We compare CAPO with no few-shot included to the default CAPO and EvoPromptGA.

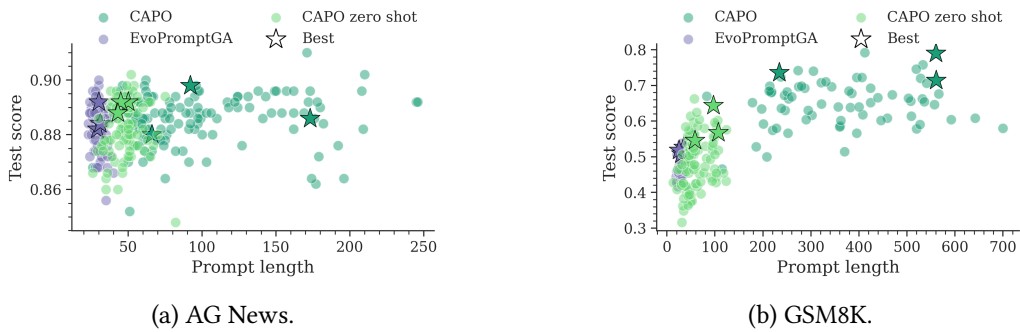

(a) AG News.

(b) GSM8K.

Figure 15: Test score vs. prompt length for every prompt with Llama-3.3-70B. A star marks the final selected prompt per seed (best performing from last step based on available dev scores). Prompt length includes both the number of tokens in the system prompt and (user) prompt. We compare CAPO with no few-shot included to the default CAPO and EvoPromptGA.

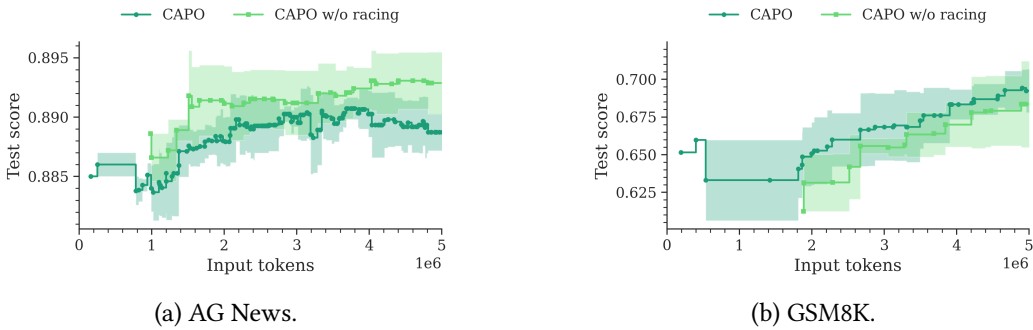

(a) AG News.

(b) GSM8K.

Figure 16: Population mean test scores over input tokens with Llama-3.3-70B. We compare CAPO without racing (one block with $b = |\mathcal{D}_{\text{dev}}|$) with the default CAPO.

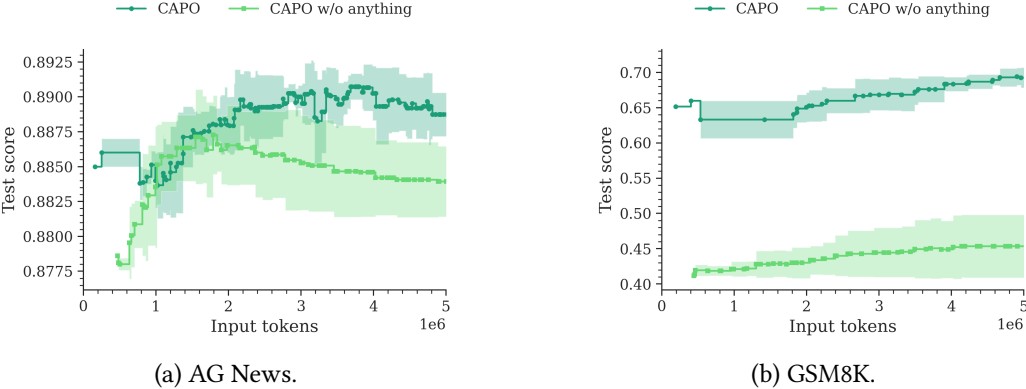

(a) AG News.

(b) GSM8K.

Figure 17: Popultation mean test scores over input tokens with Llama-3.3-70B. We compare CAPO without racing, without few-shot examples, and without length penalty to default CAPO.

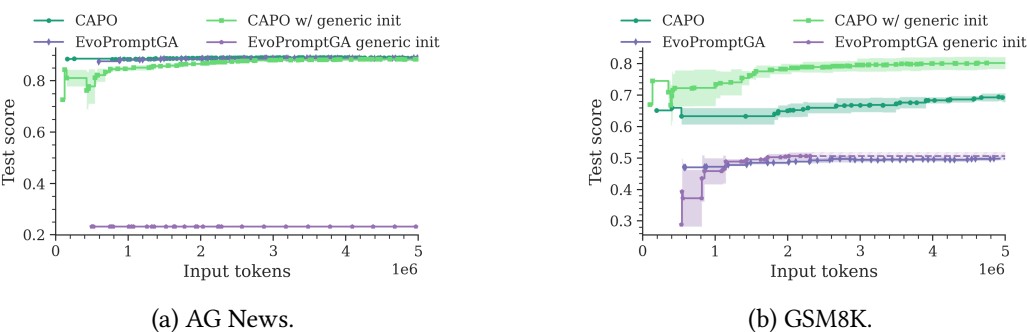

(a) AG News.

(b) GSM8K.

Figure 18: Population mean test scores over input tokens with Llama-3.3-70B. CAPO and EvoPromptGA started with generic, task-unspecific prompts.

## I.2 Impact of Racing

In Figure 19, we compare the required input token budget per step for CAPO (w/ racing), CAPO w/o racing, and EvoPromptGA on AG News with Llama-3.3-70B. All three optimizers require a large number of tokens in the first step. This is due to the additional evaluation of initial prompts on top of the candidates of the first step. Both EvoPromptGA and CAPO w/o racing remain at a constant rate afterwards. While CAPO w/o racing benefits from the prompt-evaluation-cache but suffers from long prompts potentially including few-shots, EvoPrompt has short prompts but no cache. Both effects seem to cancel out and the required input tokens stay at a constant rate of about 250k input tokens per step, allowing for roughly 19 optimization steps. In contrast, the CAPO budget requirement is already low at the beginning, as it does not necessarily need to evaluate the candidates on the entire dev set, terminating poor candidates early through racing. The required budget decreases further after 3 steps and stays roughly constant with small fluctuations around 100k tokens per step, allowing for over 70 steps with the same budget. These observations underscore the benefits of racing in terms of cost-efficiency.

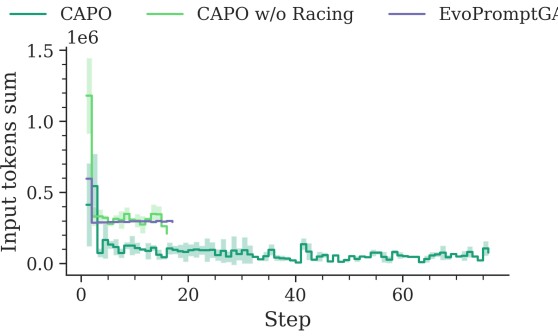

Figure 19: Sum of input tokens required per optimization step of Llama-3.3-70B on AG News. Mean and standard deviations are computed across seeds. We compare default CAPO, EvoPromptGA and CAPO without racing.

This conclusion is further supported by Table 16, where we compare the actual block evaluations required for CAPO with racing to the theoretical evaluations required if each prompt had been evaluated on all blocks. In the example of Figure 19, we save around 50% of evaluations. On average, we save 44% of evaluations over all datasets and models.

Table 16: Evaluated blocks per model and dataset with racing vs. number of required blocks that would have been required if prompts had been evaluated across all 10 blocks, averaged over seeds. We calculate how many blocks (in %) were saved by using racing.

| Dataset | Model | w/ racing | w/o racing | savings (%) |
|---|---|---|---|---|
| AG News | Llama-3.3-70B | 929.0 | 1886.7 | 50.76 |
| | Mistral-Small-24B | 608.3 | 1356.7 | 55.16 |
| | Qwen2.5-32B | 707.0 | 1310.0 | 46.03 |
| COPA | Llama-3.3-70B | 804.7 | 1690.0 | 52.39 |
| | Mistral-Small-24B | 754.7 | 1273.3 | 40.73 |
| | Qwen2.5-32B | 948.7 | 1566.7 | 39.45 |
| GSM8K | Llama-3.3-70B | 317.7 | 630.0 | 49.58 |
| | Mistral-Small-24B | 314.0 | 456.7 | 31.24 |
| | Qwen2.5-32B | 376.7 | 633.3 | 40.53 |
| SST-5 | Llama-3.3-70B | 832.7 | 1316.7 | 36.76 |
| | Mistral-Small-24B | 703.3 | 1093.3 | 35.67 |
| | Qwen2.5-32B | 836.3 | 1070.0 | 21.84 |
| Subj | Llama-3.3-70B | 648.3 | 1566.7 | 58.62 |
| | Mistral-Small-24B | 625.0 | 1260.0 | 50.40 |
| | Qwen2.5-32B | 672.7 | 1360.0 | 50.54 |
| **Avg.** | | 671.9 | 1231.3 | 43.98 |

### I.3 Influence of Meta-Prompt Simplification and Task Descriptions

To investigate the influence of our meta-prompt simplification, we perform an additional experiment with EvoPromptGA using our simplified CAPO meta-prompts, including a task description. Since EvoPromptGA uses only a single meta-prompt and LLM call to perform both cross-over and mutation, we combine our CAPO cross-over and mutation prompt into a single meta-prompt. For details, we refer to Appendix D.3. In Figure 20, we compare optimization curves for standard EvoPromptGA and EvoPromptGA with our simplified template. We observe that performance with our simplified template is slightly worse compared to the original template. Nonetheless, it is important to mention that our templates are substantially shorter in terms of number of tokens. Thus, this experiment indicates that the choice of the meta-prompt template is also a trade-off between performance and cost.

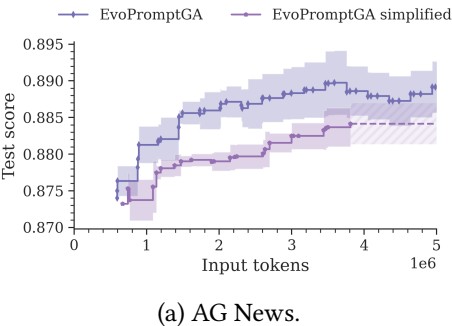

(a) AG News.

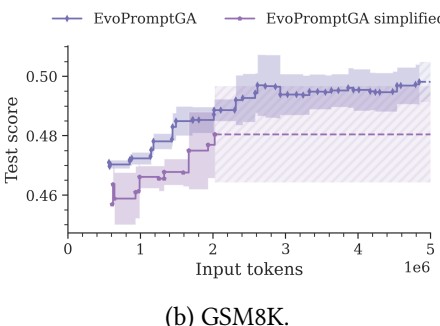

(b) GSM8K.

Figure 20: Population mean test scores over input tokens with Llama-3.3-70B. Mean and standard deviations are computed across seeds. We compare the performance of EvoPromptGA with default meta-prompts (Guo et al., 2024) to EvoPromptGA with our combined CAPO meta-prompts.

## J Best Prompts per Tasks

In the following, we report the best prompts per optimizer with Llama-3.3-70B for each dataset. The displayed prompts yield the best test-set performance across all seeds. Note that this section serves primarily to provide illustrative insights and examples of generated prompts rather than to report performance metrics.

### J.1 Initial Prompts

Table 17: Best initial prompts by test scores with Llama-3.3-70B and three exemplary generic prompts. For a full list of all initial prompts, we refer to our research repository.

---

**AG News** (88.6%):
Read the following news text and determine which category it belongs to. Choose from: World, Sports, Business, or Sci/Tech. Your final answer must be enclosed in `<final_answer> </final_answer>` tags for automated extraction.

---

**COPA** (99.2%):
Select the statement that represents the most reasonable causal relationship to the given context. Respond with `<final_answer>A</final_answer>` or `<final_answer>B</final_answer>` only.

---

**GSM8K** (52.2%):
I'm struggling with this math word problem that needs multiple steps to solve. Can you help? Make sure to put your final answer between `<final_answer> </final_answer>` tags so I can easily find it.

---

**SST-5** (60.4%):
Movie review sentiment classification task: From the following five options - very negative, negative, neutral, positive, or very positive - which best describes this review? Your answer must appear between `<final_answer>` and `</final_answer>` markers.

---

**Subj** (70.0%):
Evaluate this sentence and determine if it's presenting objective information (facts that can be verified) or subjective content (opinions, judgments, or emotions). Provide your classification inside `<final_answer> </final_answer>` markers.

---

**Generic Prompt**
Let's think step by step. Your answer should be enclosed within `<final_answer> </final_answer>` tags.

---

**Generic Prompt**
Give me your response within `<final_answer>` tags.

---

**Generic Prompt**
Please provide a thoughtful answer to my question and wrap your response in `<final_answer>` tags so I can easily identify it.

---

## J.2 CAPO Prompts

Table 18: Best prompts of CAPO by test scores, optimized and evaluated with Llama-3.3-70B.

**AG News** (91.0%):
We have a collection of news stories that need to be sorted into categories. Your task is to read the provided article and determine whether it falls under the category of World, Sports, Business, or Sci/Tech news. Once you've made your decision, please enclose your chosen category in <final_answer>answer</final_answer> tags for easy identification. *+2 few shots*

**COPA** (99.8%):
To evaluate your ability to reason about cause-and-effect relationships, this task presents you with a scenario and asks you to identify the most plausible consequence or antecedent. Given a premise, assess the two provided options, labeled A and B, and select the one that logically follows or precedes the premise, responding with either <final_answer>A</final_answer> or <final_answer>B</final_answer> to indicate your choice. *+2 few shots*

**GSM8K** (79.2%):
To tackle this math word problem, which demands a series of logical steps, dissect it methodically. Outline your thought process and ensure you clearly signify your solution, enclosing it within <final_answer> </final_answer> markers for easy identification. *+2 few shots*

**SST-5** (63.6%):
Assess the emotional tone conveyed in the provided movie review, then categorize it into one of five sentiment levels: very negative, negative, neutral, positive, or very positive, and encapsulate your chosen category within <final_answer> </final_answer> tags, following this format: <final_answer> selected_sentiment </final_answer>, to clearly denote the sentiment classification of the film review. *+2 few shots*

**Subj** (94.6%):
Label each sentence as either a statement of fact that can be proven or disproven, or a reflection of personal feelings, opinions, or biases, by categorizing it as <final_answer>objective</final_answer> if it contains information that can be verified, or <final_answer>subjective</final_answer> if it expresses emotions, attitudes, or individual evaluations, and respond with one of these two classifications. *+4 few shots*

## J.3 EvoPromptGA Prompts

Table 19: Best prompts of EvoPromptGA by test scores, optimized and evaluated with Llama-3.3-70B.

**AG News** (90.0%):
Categorize the given news article into its relevant category (World, Sports, Business, or Sci/Tech) and provide your classified response within <final_answer> tags for easy identification.

**COPA** (99.4%):
Use commonsense knowledge to identify the causally related option (A or B) to the given statement and respond with <final_answer>A</final_answer> or <final_answer>B</final_answer>.

**GSM8K** (53.8%):
Assist with solving the elementary or grade school level math problem that requires multiple steps and provide the solution within <final_answer> </final_answer> tags for easy identification.

**SST-5** (63.0%):
Evaluate the sentiment of the given movie review and categorize it as very negative, negative, neutral, positive, or very positive, enclosing the chosen category within <final_answer> and </final_answer> tags.

**Subj** (78.8%):
Determine the subjectivity or objectivity of a sentence and provide the assessment enclosed in <final_answer> tags.

### J.4 OPRO Prompts

Table 20: Best prompts of OPRO by test scores, optimized and evaluated with Llama-3.3-70B.

**AG News** (89.4%):
Classify the news article into one of four categories (World, Sports, Business, Sci/Tech) based on its content, and provide your answer in lowercase within `<final_answer>` tags for efficient data extraction and analysis, ensuring accuracy and consistency in categorization, and enabling informed decision-making with a standardized format for optimal processing and evaluation.

**COPA** (99.2%):
Select the statement that represents the most reasonable causal relationship to the given context. Respond with `<final_answer>A</final_answer>` or `<final_answer>B</final_answer>` only.

**GSM8K** (56.0%):
To solve the math problem, provide a concise, logical, and step-by-step explanation that directly addresses the problem, incorporating all necessary calculations and formulas. Ensure your reasoning is easy to follow and free of unnecessary information. Clearly present your final numerical answer within `<final_answer>` and `</final_answer>` tags, allowing for effortless identification and verification of the solution. Utilize a well-structured approach that effectively communicates the problem's resolution, enabling efficient understanding and validation of the mathematical solution.

**SST-5** (63.0%):
Analyze the movie review's sentiment by identifying the emotional tone and language used, then categorize it as very negative, negative, neutral, positive, or very positive, and provide your answer in the format: `<final_answer>` sentiment_category `</final_answer>`, considering the context, tone, and emotional cues to accurately reflect the reviewer's opinion in a concise and nuanced manner, ensuring your classification is informed by both the explicit and implicit emotional expressions in the review.

**Subj** (75.4%):
Classify the sentence as 'objective' if it presents factual information or 'subjective' if it expresses personal opinions, emotions, or biases, and provide your answer between `<final_answer>` tags, considering the sentence's content, tone, and purpose to inform a clear and accurate judgment.

### J.5 PromptWizard Prompts

We only report a single prompt optimized by PromptWizard, including system prompt for Llama-3.3-70B on AG News for the sake of conciseness and refer the interested reader to our research repository.

Table 21: Best prompt of PromptWizard by test scores, optimized and evaluated with Llama-3.3-70B.

**AG News** (23.6%)
*system prompt*:
You are a natural language processing (NLP) specialist with expertise in text classification and machine learning. You have extensive experience in developing and training models to categorize text into predefined categories. Your knowledge of NLP techniques, such as tokenization, stemming, and named entity recognition, enables you to extract relevant features from the news articles and classify them accurately. You are familiar with various machine learning algorithms, including supervised and unsupervised learning methods, and can select the most suitable approach for this task. With your expertise, you can analyze the dataset, identify the key characteristics of each category, and develop a robust classification model that can accurately assign each news article to one of the four categories: World, Sports, Business, or Sci/Tech. Your goal is to achieve high accuracy in classification, and you can use techniques such as cross-validation and hyperparameter tuning to optimize the performance of the model. By extracting the class between the markers `<final_answer>answer</final_answer>`, you can provide a clear and concise output that indicates the predicted category for each news article.
*user prompt:*
What are the key assumptions underlying this news article classification task? To simplify the problem, let's start by identifying the categories: World, Sports, Business, and Sci/Tech. How can we make progress on this problem? By reading the news article and trying to classify it into one of the four categories, we can start making progress. Let's make a list of ideas for solving this problem and apply them one by one to see if any progress can be made. Place your classification within `<final_answer>` tags. *+2 few shots*

