# OpenReview forum: "CAPO: Cost-Aware Prompt Optimization"
_automl.cc/AutoML/2025/Methods_Track — AutoML 2025 Methods Track_

### Official Review · Reviewer_UZ7P · 2025-04-29

**Comments To Authors:**

This paper introduces CAPO (Cost-Aware Prompt Optimization), a discrete prompt optimization algorithm designed to improve the cost-efficiency of finding effective prompts for large language models (LLMs). The core idea is to integrate established AutoML techniques, specifically racing (to reduce evaluations) and multi-objective optimization (via a prompt length penalty), into an evolutionary framework (based on EvoPromptGA) that uses a Meta-LLM and Meta-Prompts as operators. CAPO aims to optimize both the instruction and few-shot examples simultaneously, leveraging task descriptions for robustness against initial prompt quality.

The work addresses the important and practical problem of high computational cost in prompt optimization, which is a significant barrier to wider adoption. Integrating AutoML techniques like racing and multi-objective optimization into this context is a valuable contribution and could stimulate further research at the intersection of AutoML and LLM prompting. The joint optimization of instructions and examples, along with the use of task descriptions for robustness, are also positive aspects.

I believe that the paper presents a promising direction for cost-efficient prompt optimization by integrating valuable AutoML techniques. The methodology is generally sound, and the initial results are encouraging. The strength of the empirical evidence is currently limited by the lack of statistically significant tests. However, this is acceptable given the computational cost of the tests. Finally, the limitations are clearly pointed out by the authors in the conclusion.

After my review, I am inclined to confidently recommend that the paper be accepted.

**Review Confidence:**

4

**Review Rating:**

8

---

### Official Review · Reviewer_UySb · 2025-04-30

**Comments To Authors:**

&nbsp;

**__SUMMARY__**

&nbsp;

The authors introduce cost aware prompt optimization (CAPO), a novel algorithm incorporating concepts from racing algorithms and multi-objective optimization to improve the cost efficiency of prompt optimization. The paper empirical evaluation appears to be solid and the authors have provided a well-documented codebase ensuring full reproducibility of the work. Furthermore the paper is very is very well-written and presented. A such I recommend acceptance with the following points the authors may wish to consider.

&nbsp;

**__MAJOR POINTS__**

&nbsp;

1. Given that the main motivation of the current work is to minimize fiscal cost, it may be nice to provide an actual $ cost comparison in addition to comparison based on input tokens.

2. For the few-shot examples why is an evaluation LLM used to label the few-shot examples? Why not use the ground truth label? Additionally what is reasoning and why is it required?

3. In Algorithm 2 in Appendix B, Line 4, are the two prompts selected for crossover sampled with or without replacement?

4. In Algorithm 2 in Appendix B, what is the purpose of Line 16 in the mutation function? In other words what is the purpose of creating new few-shot examples at random?

5. In the related work on prompt optimization it would be worth discussing frameworks such as Aviary [7], TextGrad [8], and DSPy [9] which seek to optimize black-box prompts via gradients.

&nbsp;

**__MINOR POINTS__**

&nbsp;

1. On line 15, "21%" would be clearer.

2. There are some missing capitalizations in the references e.g. "ChatGPT" and "LLMs".

3. The arXiv link is missing for the Hoang et al. reference.

4. In the introduction when citing in-context learning, [1] would be a more appropriate reference.

5. Line 771 of the appendix, sentence appears to need a revision.

6. In section A.1 of the appendix it may be worth mentioning the term black-box prompt optimization [2] as an alternative term for discrete prompt optimization. It may also be worth mentioning the works [2-5] for prompt optimization and the related work [6] which considers cost-aware inference hyperparameter optimization.

7. Line 64, typo, "as an additional objective".

8. In Equation 1, it is generally bad practice to include a full stop at the end of a case. It may be better to restructure the sentence to avoid this.

9. Line 112, typo, "mispredicted".

10. Line 788, typo, "fewer evaluations". "Less" is used with continuous quantities such as time whereas "fewer" is used with discrete quantities such as evaluations.

11. Line 828, typo, "categorized into".

12. In Algorithm 2, perhaps bolded "or" would be clearer than the carrot symbol to denote the logical OR operation?

13. Line 246, it may be worth reminding the reader that DE stands for directed evolution?

14. In Table 2, what is the column with the "empty set" symbol as the heading? Update: From reading Table 16 in the appendix it appears as though it is the average. It would be worth stating this in the table caption.

15. In footnote 5, the authors state that there is no clear way to increase the compute time of PromptWizard and as such, the authors report its performance on a reduced budget. The budget is in terms of input tokens? How many input tokens is the reduced budget? Does this constitute a fair comparison? How does this related to the curve for PromptWizard in Figure 2?

16. Line 933, typo, "fewer input tokens".

17. In Section J.1 of the appendix there is a missing full stop at the end of Equation 3.

18. Section A.1 appears to be largely repeated relative to Section 3 in the main paper?

&nbsp;


**__REFERENCES__**

&nbsp;

[1] Brown, T., Mann, B., Ryder, N., Subbiah, M., Kaplan, J.D., Dhariwal, P., Neelakantan, A., Shyam, P., Sastry, G., Askell, A. and Agarwal, S., 2020. [Language models are few-shot learners.](https://proceedings.neurips.cc/paper/2020/hash/1457c0d6bfcb4967418bfb8ac142f64a-Abstract.html) Advances in Neural Information Processing Systems, 33, pp.1877-1901.

[2] Cheng, J., Liu, X., Zheng, K., Ke, P., Wang, H., Dong, Y., Tang, J. and Huang, M., 2024, August. [Black-Box Prompt Optimization: Aligning Large Language Models without Model Training.](https://aclanthology.org/2024.acl-long.176/) In Proceedings of the 62nd Annual Meeting of the Association for Computational Linguistics (Volume 1: Long Papers) (pp. 3201-3219).

[3] Lin, X., Dai, Z., Verma, A., Ng, S.K., Jaillet, P. and Low, B.K.H., 2024. [Prompt optimization with human feedback.](https://arxiv.org/abs/2405.17346) arXiv preprint arXiv:2405.17346.

[4] Wu, Z., Lin, X., Dai, Z., Hu, W., Shu, Y., Ng, S.K., Jaillet, P. and Low, B.K.H., [Prompt Optimization with EASE? Efficient Ordering-aware Automated Selection of Exemplars.](https://openreview.net/forum?id=6uRrwWhZlM) In The Thirty-eighth Annual Conference on Neural Information Processing Systems.

[5] Hu, W., Shu, Y., Yu, Z., Wu, Z., Lin, X., Dai, Z., Ng, S.K. and Low, B.K.H., 2024. [Localized zeroth-order prompt optimization.](https://proceedings.neurips.cc/paper_files/paper/2024/hash/9cef1316eaef9bd99da46f63334dc031-Abstract-Conference.html) Advances in Neural Information Processing Systems, 37, pp.86309-86345.

[6] Wang, C., Liu, X. and Awadallah, A.H., 2023, December. [Cost-effective hyperparameter optimization for large language model generation inference.](https://proceedings.mlr.press/v224/wang23b.html) In International Conference on Automated Machine Learning (pp. 21-1). PMLR.

[7] Narayanan, S., Braza, J.D., Griffiths, R.R., Ponnapati, M., Bou, A., Laurent, J., Kabeli, O., Wellawatte, G., Cox, S., Rodriques, S.G. and White, A.D., 2024. [Aviary: training language agents on challenging scientific tasks.](https://arxiv.org/abs/2412.21154) arXiv preprint arXiv:2412.21154.

[8] Yuksekgonul, M., Bianchi, F., Boen, J., Liu, S., Lu, P., Huang, Z., Guestrin, C. and Zou, J., 2025. [Optimizing generative AI by backpropagating language model feedback.](https://www.nature.com/articles/s41586-025-08661-4) Nature, 639(8055), pp.609-616.

[9] Khattab, O., Singhvi, A., Maheshwari, P., Zhang, Z., Santhanam, K., Haq, S., Sharma, A., Joshi, T.T., Moazam, H., Miller, H. and Zaharia, M., 2024. [DSPy: Compiling declarative language model calls into state-of-the-art pipelines.](https://openreview.net/forum?id=sY5N0zY5Od) In The Twelfth International Conference on Learning Representations.

&nbsp;

**Review Confidence:**

5

**Review Rating:**

9

---

### Official Review · Reviewer_DciR · 2025-05-02

**Comments To Authors:**

This paper proposes a cost-aware discrete prompt optimization method based on the previous AutoML techniques. The proposed method, termed the cost-aware prompt optimization (CAPO), leverages the racing mechanism to reduce the API calls of LLMs and injects the penalty term into the objective function to balance the cost and performance. The cost reduction techniques are integrated into EvoPrompt. The experimental evaluation demonstrates that the proposed CAPO outperforms the several existing discrete prompt optimization methods and achieves a reduction in the evaluations and prompt length.

The prompt optimization is a promising application of AutoML in the LLM domain. The cost-aware approach itself has been investigated in the AutoML community, but its application to discrete prompt optimization will be novel. This paper is easy to follow, and the experimental results show the effectiveness of CAPO.

The following is my concern about the proposed CAPO:

1.
In the prompt optimization, two LLMs, meta-LLM and evaluation-LLM, exist. The authors use the same LLM for meta-LLM and evaluation-LLM in each experimental setting. The racing mechanism only reduces the number of API calls for the evaluation-LLM but not for the meta-LLM. In some cases, the meta-LLM can be more expensive than the evaluation LLM if different LLMs are used when optimizing the prompt for smaller models. It might be interesting to check the breakdown of the token consumption for the meta-LLM and evaluation-LLM in the experiments.

2.
The proposed method includes several additional techniques compared to the baseline EvoPrompt. Specifically, the reviewer supposes that the optimization of few-shot examples does not aim to reduce the cost but to improve the performance. The reviewer would suggest that the authors include the performance of CAPO without few-shot example optimization (i.e., zero-shot) and/or CAPO with no racing and $\gamma=0$ in Table 2 for detailed comparison.


[Strengths]
- A promising application of cost reduction techniques in AutoML to discrete prompt optimization.
- The experimental results demonstrate that the proposed CAPO can reduce the token length and archive higher scores with small token consumptions.

[Weaknesses]
- The techniques, the racing, token penalty, and optimization of few-shot examples are based on the previous works. The technical novelty of the paper might be limited.

[Minor comments]
- In Table 2, what does "$\varnothing$" mean in the rightest column?

**Review Confidence:**

4

**Review Rating:**

7

---

### Meta-Review · Area_Chair_UCt3 · 2025-05-11

**Recommendation:** Accept
**Confidence:** 4

**Metareview:**

This paper proposes an AutoML-based cost-aware method called CAPO for discrete prompt optimization incoporating racing and multi-objective optimization.

This paper has several strengths. First, the application of cost-aware methods to discrete prompt optimization is novel. Second, the method has potential for significant impact due to the computational cost of prompt optimization in practice. Third, empirical results show that CAPO achieves greater performance with fewer tokens than baseline approaches. Moreover, the paper is well-written and the codebase is reproducible.

Some reviewers expressed concerns about various aspects of the paper. First, the method does not fully consider the computational cost of the meta-LLM. Second, although CAPO is novel in exploiting the intersection of cost-aware methods and discrete prompt optimization, the technical novelty of the method itself is limited. Third, some related work on prompt optimization is missing.

Overall, this is a solid paper with a good mix of novelty, significance, and empirical results. I therefore recommend acceptance.